# V-MPO: On-Policy Maximum a Posteriori Policy Optimization for Discrete and Continuous Control

**H. Francis Song,**[*] **Abbas Abdolmaleki,**[*] **Jost Tobias Springenberg, Aidan Clark, Hubert Soyer, Jack W. Rae, Seb Noury, Arun Ahuja, Siqi Liu, Dhruva Tirumala, Nicolas Heess, Dan Belov, Martin Riedmiller, Matthew M. Botvinick**
DeepMind, London, UK
```
{songf,aabdolmaleki,springenberg,aidanclark,
 soyer,jwrae,snoury,arahuja,liusiqi,dhruvat,
 heess,danbelov,riedmiller,botvinick}@google.com
```

## Abstract

Some of the most successful applications of deep reinforcement learning to challenging domains in discrete and continuous control have used policy gradient methods in the on-policy setting. However, policy gradients can suffer from large variance that may limit performance, and in practice require carefully tuned entropy regularization to prevent policy collapse. As an alternative to policy gradient algorithms, we introduce V-MPO, an on-policy adaptation of Maximum a Posteriori Policy Optimization (MPO) that performs policy iteration based on a learned state-value function. We show that V-MPO surpasses previously reported scores for both the Atari-57 and DMLab-30 benchmark suites in the multi-task setting, and does so reliably without importance weighting, entropy regularization, or population-based tuning of hyperparameters. On individual DMLab and Atari levels, the proposed algorithm can achieve scores that are substantially higher than has previously been reported. V-MPO is also applicable to problems with high-dimensional, continuous action spaces, which we demonstrate in the context of learning to control simulated humanoids with 22 degrees of freedom from full state observations and 56 degrees of freedom from pixel observations, as well as example OpenAI Gym tasks where V-MPO achieves substantially higher asymptotic scores than previously reported.

## 1 Introduction

Deep reinforcement learning (RL) with neural network function approximators has achieved superhuman performance in several challenging domains (Mnih et al., 2015; Silver et al., 2016; 2018). Some of the most successful recent applications of deep RL to difficult environments such as Dota 2 (OpenAI, 2018a), Capture the Flag (Jaderberg et al., 2019), Starcraft II (Vinyals et al., 2019), and dexterous object manipulation (OpenAI, 2018b) have used policy gradient-based methods such as Proximal Policy Optimization (PPO) (Schulman et al., 2017) and the Importance-Weighted Actor-Learner Architecture (IMPALA) (Espeholt et al., 2018), both in the approximately on-policy setting.

Policy gradients, however, can suffer from large variance that may limit performance, especially for high-dimensional action spaces (Wu et al., 2018). In practice, moreover, policy gradient methods typically employ carefully tuned entropy regularization in order to prevent policy collapse. As an alternative to policy gradient-based algorithms, in this work we introduce an approximate policy iteration algorithm that adapts Maximum a Posteriori Policy Optimization (MPO) (Abdolmaleki et al., 2018a;b) to the on-policy setting. The modified algorithm, V-MPO, relies on a learned state-value function $V(s)$ instead of the state-action value function used in MPO. Like MPO, rather than directly updating the parameters in the direction of the policy gradient, V-MPO first constructs a target distribution for the policy update subject to a sample-based KL constraint, then calculates the gradient that partially moves the parameters toward that target, again subject to a KL constraint.

---

[*]Equal contribution

As we are particularly interested in scalable RL algorithms that can be applied to multi-task settings where a single agent must perform a wide variety of tasks, we show for the case of discrete actions that the proposed algorithm surpasses previously reported performance in the multi-task setting for both the Atari-57 (Bellemare et al., 2012) and DMLab-30 (Beattie et al., 2016) benchmark suites, and does so reliably without population-based tuning of hyperparameters (Jaderberg et al., 2017a). For a few individual levels in DMLab and Atari we also show that V-MPO can achieve scores that are substantially higher than has previously been reported in the single-task setting, especially in the challenging Ms. Pacman.

V-MPO is also applicable to problems with high-dimensional, continuous action spaces. We demonstrate this in the context of learning to control both a 22-dimensional simulated humanoid from full state observations—where V-MPO reliably achieves higher asymptotic performance than previous algorithms—and a 56-dimensional simulated humanoid from pixel observations (Tassa et al., 2018; Merel et al., 2019). In addition, for several OpenAI Gym tasks (Brockman et al., 2016) we show that V-MPO achieves higher asymptotic performance than has previously been reported.

## 2 BACKGROUND AND SETTING

We consider the discounted RL setting, where we seek to optimize a policy $\pi$ for a Markov Decision Process described by states $s$, actions $a$, initial state distribution $\rho_0^{\text{env}}(s_0)$, transition probabilities $\mathcal{P}^{\text{env}}(s_{t+1}|s_t, a_t)$, reward function $r(s_t, a_t)$, and discount factor $\gamma \in (0, 1)$. In deep RL, the policy $\pi_\theta(a_t|s_t)$, which specifies the probability that the agent takes action $a_t$ in state $s_t$ at time $t$, is described by a neural network with parameters $\theta$. We consider problems where both the states $s$ and actions $a$ may be discrete or continuous. Two functions play a central role in RL: the state-value function $V^\pi(s_t) = \mathbb{E}_{a_t, s_{t+1}, a_{t+1}, \ldots} \left[ \sum_{k=0}^\infty \gamma^k r(s_{t+k}, a_{t+k}) \right]$ and the state-action value function $Q^\pi(s_t, a_t) = \mathbb{E}_{s_{t+1}, a_{t+1}, \ldots} \left[ \sum_{k=0}^\infty \gamma^k r(s_{t+k}, a_{t+k}) \right] = r(s_t, a_t) + \gamma \mathbb{E}_{s_{t+1}} \left[ V^\pi(s_{t+1}) \right]$, where $s_0 \sim \rho_0^{\text{env}}(s_0)$, $a_t \sim \pi(a_t|s_t)$, and $s_{t+1} \sim \mathcal{P}^{\text{env}}(s_{t+1}|s_t, a_t)$.

In the usual formulation of the RL problem, the goal is to find a policy $\pi$ that maximizes the expected return given by $J(\pi) = \mathbb{E}_{s_0, a_0, s_1, a_1, \ldots} \left[ \sum_{t=0}^\infty \gamma^t r(s_t, a_t) \right]$. In policy gradient algorithms (Williams, 1992; Sutton et al., 2000; Mnih et al., 2016), for example, this objective is directly optimized by estimating the gradient of the expected return. An alternative approach to finding optimal policies derives from research that treats RL as a problem in probabilistic inference, including Maximum a Posteriori Policy Optimization (MPO) (Levine, 2018; Abdolmaleki et al., 2018a;b). Here our objective is subtly different, namely, given a suitable criterion for what are good actions to take in a certain state, how do we find a policy that achieves this goal?

As was the case for the original MPO algorithm, the following derivation is valid for any such criterion. However, the *policy improvement theorem* (Sutton & Barto, 1998) tells us that a policy update performed by exact policy iteration, $\pi(s) = \arg\max_a [Q^\pi(s, a) - V^\pi(s)]$, can improve the policy if there is at least one state-action pair with a positive advantage and nonzero probability of visiting the state. Motivated by this classic result, in this work we specifically choose an exponential function of the advantages $A^\pi(s, a) = Q^\pi(s, a) - V^\pi(s)$.

*Notation.* In the following we use $\sum_{s,a}$ to indicate both discrete and continuous sums (i.e., integrals) over states $s$ and actions $a$ depending on the setting. A sum with indices only, such as $\sum_{s,a}$, denotes a sum over all possible states and actions, while $\sum_{s,a \sim \mathcal{D}}$, for example, denotes a sum over sample states and actions from a batch of trajectories (the "dataset") $\mathcal{D}$.

## 3 RELATED WORK

V-MPO shares many similarities, and thus relevant related work, with the original MPO algorithm (Abdolmaleki et al., 2018a;b). In particular, the general idea of using KL constraints to limit the size of policy updates is present in both Trust Region Policy Optimization (TRPO; Schulman et al., 2015) and Proximal Policy Optimization (PPO) (Schulman et al., 2017); we note, however, that this corresponds to the E-step constraint in V-MPO.

It is worth noting here the following main differences with MPO, which is conceptually quite similar to V-MPO. MPO is primarily designed to be a sample-efficient off-policy algorithm in which the

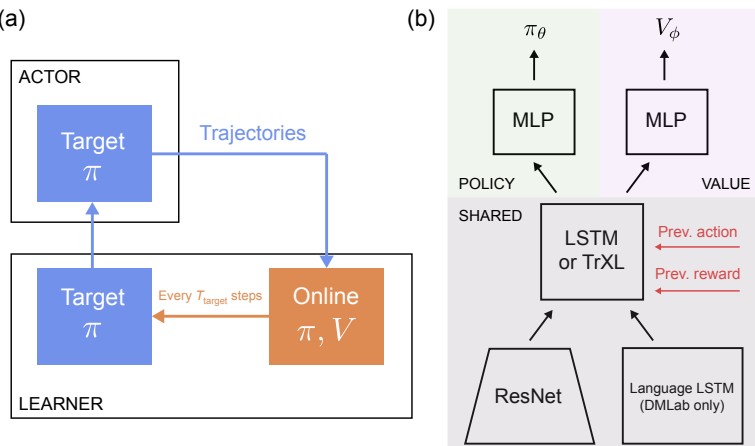

Figure 1: (a) Actor-learner architecture with a target network, which is used to generate agent experience in the environment and is updated every $T_{\mathrm{target}}$ learning steps from the online network. (b) Schematic of the agents, with the policy ($\theta$) and value ($\phi$) networks sharing most of their parameters through a shared input encoder and LSTM [or Transformer-XL (TrXL) for single Atari levels]. The agent also receives the action and reward from the previous step as an input to the LSTM. For DMLab an additional LSTM is used to process simple language instructions.

E-step constructs a conditional target distribution $q(a|s)$, which requires a state-action value function $Q(s, a)$ that can evaluate multiple sampled actions for a given state. In contrast, V-MPO is primarily (though not exclusively) designed to be an on-policy algorithm in which the E-step constructs a joint distribution $\psi(s, a)$, and in the absence of a learned $Q$-function only one action per state is used. In this regard V-MPO can also be compared to Fitted $Q$-iteration by Advantage Weighted Regression (Neumann & Peters, 2009), which learns a $Q$-function but uses only one action per state.

V-MPO can also be related to Relative Entropy Policy Search (REPS) (Peters et al., 2008). Two distinguishing features of V-MPO from REPS are the introduction of the M-step KL constraint and the use of top-$k$ advantages. Moreover, in REPS the value function is a linear function of a learned feature representation whose parameters are trained by matching the feature distributions under the policy's stationary state distribution. In V-MPO, the nonlinear neural network value function is instead learned directly from $n$-step returns. Interestingly, previous attempts to use REPS with neural network function approximators reported very poor performance, being particularly prone to local optima (Duan et al., 2016). In contrast, we find that the principles of EM-style policy optimization, when combined with this learned value function and appropriate constraints, can reliably train powerful neural networks, including transformers, for RL tasks.

Like V-MPO, Supervised Policy Update (SPU) (Vuong et al., 2019) seeks to exactly solve an optimization problem and fit the parametric policy to this solution. As we argue in Appendix D, however, SPU uses this nonparametric distribution quite differently from V-MPO; as a result, the final algorithm is closer to a policy gradient algorithm such as PPO.

## 4 METHOD

V-MPO is an approximate policy iteration (Sutton & Barto, 1998) algorithm with a specific prescription for the policy improvement step. In general, policy iteration uses the fact that the true state-value function $V^\pi$ corresponding to policy $\pi$ can be used to obtain an improved policy $\pi'$. Thus we can

1. Generate trajectories $\tau$ from an old policy $\pi_{\theta_{\mathrm{old}}}(a|s)$ whose parameters $\theta_{\mathrm{old}}$ are fixed. To control the amount of data generated by a particular policy, we use a target network which is fixed for $T_{\mathrm{target}}$ learning steps (Fig. 1a).

2. Evaluate the policy $\pi_{\theta_{\text{old}}}(a|s)$ by learning the value function $V^{\pi_{\theta_{\text{old}}}}(s)$ from empirical returns and estimating the corresponding advantages $A^{\pi_{\theta_{\text{old}}}}(s, a)$ for the actions that were taken (Section 4.1).

3. Based on $A^{\pi_{\theta_{\text{old}}}}(s, a)$, estimate an improved policy $\pi_\theta(a|s)$ which we call the "online" policy to distinguish it from the fixed target network (Section 4.2).

The first two steps are standard, and describing V-MPO's approach to step 3 is the essential contribution of this work. At a high level, our strategy is to first construct a nonparametric target distribution for the policy update, then partially move the parametric policy towards this distribution subject to a KL constraint. We first review policy evaluation (step 2) in Section 4.1, then derive the V-MPO policy improvement (step 3) in Section 4.2. Ultimately, we use gradient descent to optimize a single, relatively simple loss, which is given in Eq. 10 following the derivation. A summary of the full algorithm is also presented in Algorithm 1.

## 4.1 POLICY EVALUATION

In the present setting, policy evaluation means learning an approximate state-value function $V^\pi(s)$ given a policy $\pi(a|s)$, which we keep fixed for $T_{\text{target}}$ learning steps (i.e., batches of trajectories). We note that the value function corresponding to the target policy is instantiated in the "online" network receiving gradient updates; bootstrapping uses the online value function, as it is the best available estimate of the value function for the target policy. Thus in this section $\pi$ refers to $\pi_{\theta_{\text{old}}}$, while the value function update is performed on the current $\phi$, which may share parameters with the current $\theta$.

We fit a parametric value function $V_\phi^\pi(s)$ with parameters $\phi$ by minimizing the squared loss

$$\mathcal{L}_V(\phi) = \frac{1}{2|\mathcal{D}|} \sum_{s_t \sim \mathcal{D}} \left( V_\phi^\pi(s_t) - G_t^{(n)} \right)^2, \tag{1}$$

where $G_t^{(n)}$ is the standard $n$-step target for the value function at state $s_t$ at time $t$ (Sutton & Barto, 1998). This return uses the actual rewards in the trajectory and bootstraps from the value function for the rest: for each $\ell = t, \ldots, t + n - 1$ in an unroll, $G_\ell^{(n)} = \sum_{k=\ell}^{t+n-1} \gamma^{k-\ell} r_k + \gamma^{t+n-\ell} V_\phi^\pi(s_{t+n})$. The advantages, which are the key quantity of interest for the policy improvement step in V-MPO, are then given by $A^\pi(s_t, a_t) = G_t^{(n)} - V_\phi^\pi(s_t)$ for each $s_t, a_t$ in the batch of trajectories.

*PopArt normalization.* As we are interested in the multi-task setting where a single agent must learn a large number of tasks with differing reward scales, we used PopArt (van Hasselt et al., 2016; Hessel et al., 2018) for the value function, even when training on a single task. We observed benefits in using PopArt even in the single-task setting, partly due to the fact that we do not tune the relative weighting of the policy evaluation and policy improvement losses despite sharing most parameters for the policy and value networks. Specifically, the value function outputs a separate value for each task in normalized space, which is converted to actual returns by a shift and scaling operation, the statistics of which are learned during training. We used a scale lower bound of $10^{-2}$, scale upper bound of $10^6$, and learning rate of $10^{-4}$ for the statistics. The lower bound guards against numerical issues when rewards are extremely sparse.

*Importance-weighting for off-policy data.* It is possible to importance-weight the samples using V-trace to correct for off-policy data (Espeholt et al., 2018), for example when data is taken from a replay buffer. For simplicity, however, no importance-weighting was used for the experiments presented in this work, which were mostly on-policy.

## 4.2 POLICY IMPROVEMENT IN V-MPO

In this section we show how, given the advantage function $A^{\pi_{\theta_{\text{old}}}}(s, a)$ for the state-action distribution $p_{\theta_{\text{old}}}(s, a) = \pi_{\theta_{\text{old}}}(a|s)p(s)$ induced by the old policy $\pi_{\theta_{\text{old}}}(a|s)$, we can estimate an improved policy $\pi_\theta(a|s)$. More formally, let $\mathcal{I}$ denote the binary event that the new policy is an improvement (in a sense to be defined below) over the previous policy: $\mathcal{I} = 1$ if the policy is successfully improved and 0 otherwise. Then we would like to find the mode of the posterior distribution over parameters $\theta$ conditioned on this event, i.e., we seek the maximum a posteriori (MAP) estimate

$$\theta^* = \arg\max_\theta \left[ \log p_\theta(\mathcal{I} = 1) + \log p(\theta) \right], \tag{2}$$

where we have written $p(\mathcal{I} = 1|\theta)$ as $p_\theta(\mathcal{I} = 1)$ to emphasize the parametric nature of the dependence on $\theta$. We use the well-known identity $\log p(X) = \mathbb{E}_{\psi(Z)}\left[\log \frac{p(X,Z)}{\psi(Z)}\right] + D_{\mathrm{KL}}\big(\psi(Z)\|p(Z|X)\big)$ for any latent distribution $\psi(Z)$, where $D_{\mathrm{KL}}(\psi(Z)\|p(Z|X))$ is the Kullback-Leibler divergence between $\psi(Z)$ and $p(Z|X)$ with respect to $Z$, and the first term is a lower bound because the KL divergence is always non-negative. Then considering $s, a$ as latent variables,

$$\log p_\theta(\mathcal{I} = 1) = \sum_{s,a} \psi(s,a) \log \frac{p_\theta(\mathcal{I} = 1, s, a)}{\psi(s,a)} + D_{\mathrm{KL}}\big(\psi(s,a)\|p_\theta(s,a|\mathcal{I} = 1)\big). \tag{3}$$

Policy improvement in V-MPO consists of the following two steps which have direct correspondences to the expectation maximization (EM) algorithm (Neal & Hinton, 1998): In the expectation (E) step, we choose the variational distribution $\psi(s, a)$ such that the lower bound on $\log p_\theta(\mathcal{I} = 1)$ is as tight as possible, by minimizing the KL term. In the maximization (M) step we then find parameters $\theta$ that maximize the corresponding lower bound, together with the prior term in Eq. 2.

### 4.2.1 E-STEP

In the E-step, our goal is to choose the variational distribution $\psi(s, a)$ such that the lower bound on $\log p_\theta(\mathcal{I} = 1)$ is as tight as possible, which is the case when the KL term in Eq. 3 is zero. Given the old parameters $\theta_{\mathrm{old}}$, this simply leads to $\psi(s, a) = p_{\theta_{\mathrm{old}}}(s, a|\mathcal{I} = 1)$, or

$$\psi(s,a) = \frac{p_{\theta_{\mathrm{old}}}(s,a)p_{\theta_{\mathrm{old}}}(\mathcal{I} = 1|s,a)}{p_{\theta_{\mathrm{old}}}(\mathcal{I} = 1)}, \qquad p_{\theta_{\mathrm{old}}}(\mathcal{I} = 1) = \sum_{s,a} p_{\theta_{\mathrm{old}}}(s,a)p_{\theta_{\mathrm{old}}}(\mathcal{I} = 1|s,a). \tag{4}$$

Intuitively, this solution weights the probability of each state-action pair with its relative improvement probability $p_{\theta_{\mathrm{old}}}(\mathcal{I} = 1|s, a)$. We now choose a distribution $p_{\theta_{\mathrm{old}}}(\mathcal{I} = 1|s, a)$ that leads to our desired outcome. As we prefer actions that lead to a higher advantage in each state, we suppose that this probability is given by

$$p_{\theta_{\mathrm{old}}}(\mathcal{I} = 1|s,a) \propto \exp\left(\frac{A^{\pi_{\theta_{\mathrm{old}}}}(s,a)}{\eta}\right) \tag{5}$$

for some temperature $\eta > 0$, from which we obtain the equation on the right in Eq. 12. This probability depends on the *old* parameters $\theta_{\mathrm{old}}$ and not on the new parameters $\theta$. Meanwhile, the value of $\eta$ allows us to control the diversity of actions that contribute to the weighting, but at the moment is arbitrary. It turns out, however, that we can tune $\eta$ as part of the optimization, which is desirable since the optimal value of $\eta$ changes across iterations. The convex loss that achieves this, Eq. 13, is derived in Appendix A by minimizing the KL term in Eq. 3 subject to a hard constraint on $\psi(s, a)$.

*Top-k advantages.* We found that learning improves substantially if we take only the samples corresponding to the highest 50% of advantages in each batch for the E-step, corresponding to the use of $\tilde{\mathcal{D}}$ rather than $\mathcal{D}$ in Eqs. 12, 13. Importantly, these must be consistent between the maximum likelihood weights in Eq. 12 and the temperature loss in Eq. 13, since, mathematically, this corresponds to a specific choice of the policy improvement probability in Eq. 5 to only use the top half of the advantages. This is similar to the technique used in the Cross Entropy Method (CEM) (Mannor et al., 2003) and Covariance Matrix Adaptation - Evolutionary Strategy (CMA-ES) (Hansen et al., 1997; Abdolmaleki et al., 2017), and is a special case of the more general feature that any rank-preserving transformation is allowed under this formalism. For example, in Fig. 8 of the Appendix we show an example of an agent trained with *uniform weights* given to the top-$k$ samples, instead of optimizing the temperature. Other choices are possible, and in future work we will investigate the suitability of different choices for specific applications.

*Importance weighting for off-policy corrections.* As for the value function, importance weights can be used in the policy improvement step to correct for off-policy data. While not used for the experiments presented in this work, details for how to carry out this correction are given in Appendix E.

### 4.2.2 M-STEP: CONSTRAINED SUPERVISED LEARNING OF THE PARAMETRIC POLICY

In the E-step we found the nonparametric variational state-action distribution $\psi(s, a)$, Eq. 4, that gives the tightest lower bound to $p_\theta(\mathcal{I} = 1)$ in Eq. 3. In the M-step we maximize this lower bound

together with the prior term $\log p(\theta)$ with respect to the parameters $\theta$, which effectively leads to a constrained weighted maximum likelihood problem. Thus the introduction of the nonparametric distribution in Eq. 4 separates the RL procedure from the neural network fitting.

We would like to find new parameters $\theta$ that minimize

$$\mathcal{L}(\theta) = -\sum_{s,a} \psi(s,a) \log \frac{p_\theta(\mathcal{I}=1, s, a)}{\psi(s,a)} - \log p(\theta). \tag{6}$$

Note, however, that so far we have worked with the joint state-action distribution $\psi(s,a)$ while we are in fact optimizing for the policy, which is the conditional distribution $\pi_\theta(a|s)$. Writing $p_\theta(s,a) = \pi_\theta(a|s)p(s)$ since only the policy is parametrized by $\theta$ and dropping terms that are not parametrized by $\theta$, the first term of Eq. 6 is seen to be the weighted maximum likelihood policy loss

$$\mathcal{L}_\pi(\theta) = -\sum_{s,a} \psi(s,a) \log \pi_\theta(a|s). \tag{7}$$

In the sample-based computation of this loss, we assume that any state-action pairs not in the batch of trajectories have zero weight, leading to the normalization in Eq. 12.

As in the original MPO algorithm, a useful prior is to keep the new policy $\pi_\theta(a|s)$ close to the old policy $\pi_{\theta_{old}}(a|s)$: $\log p(\theta) \approx -\alpha\mathbb{E}_{s\sim p(s)}\big[D_{KL}\big(\pi_{\theta_{old}}(a|s)\|\pi_\theta(a|s)\big)\big]$. While intuitive, we motivate this more formally in Appendix B. It is again more convenient to specify a bound on the KL divergence instead of tuning $\alpha$ directly, so we solve the constrained optimization problem

$$\theta^* = \arg\min_\theta -\sum_{s,a} \psi(s,a) \log \pi_\theta(a|s) \quad \text{s.t.} \quad \mathbb{E}_{s\sim p(s)}\Big[D_{KL}\big(\pi_{\theta_{old}}(a|s)\|\pi_\theta(a|s)\big)\Big] < \epsilon_\alpha. \tag{8}$$

Intuitively, the constraint in the E-step expressed by Eq. 18 in Appendix A for tuning the temperature only constrains the nonparametric distribution; it is the constraint in Eq. 8 that directly limits the change in the parametric policy, in particular for states and actions that were not in the batch of samples and which rely on the generalization capabilities of the neural network function approximator.

To make the constrained optimization problem amenable to gradient descent, we use Lagrangian relaxation to write the unconstrained objective as

$$\mathcal{J}(\theta, \alpha) = \mathcal{L}_\pi(\theta) + \alpha\bigg(\epsilon_\alpha - \mathbb{E}_{s\sim p(s)}\Big[D_{KL}\big(\pi_{\theta_{old}}(a|s)\|\pi_\theta(a|s)\big)\Big]\bigg), \tag{9}$$

which we can optimize by following a coordinate-descent strategy, alternating between the optimization over $\theta$ and $\alpha$. Since $\eta$ and $\alpha$ are Lagrange multipliers that must be positive, after each gradient update we project the resulting $\eta$ and $\alpha$ to a small positive value which we choose to be $\eta_{min} = \alpha_{min} = 10^{-8}$ throughout the results presented below.

KL constraints in both the E-step and M-step are generally well satisfied, especially for the E-step since the temperature optimization is convex. Fig. 7 in the Appendix shows an example of how the KL constraints behave in the Atari Seaquest experiment presented below. We note, in particular, that it is desirable for the bounds to not just be satisfied but *saturated*.

### 4.3 FULL LOSS FUNCTION

In this section we provide the full loss function used to implement V-MPO, which is perhaps simpler than is suggested by the derivation. Consider a batch of data $\mathcal{D}$ consisting of a number of trajectories, with $|\mathcal{D}|$ total state-action samples. Each trajectory consists of an unroll of length $n$ of the form $\tau = \big[(s_t, a_t, r_{t+1}), \ldots, (s_{t+n-1}, a_{t+n-1}, r_{t+n}), s_{t+n}\big]$ including the bootstrapped state $s_{t+n}$, where $r_{t+1} = r(s_t, a_t)$. The total loss is the sum of a policy evaluation loss and a policy improvement loss,

$$\mathcal{L}(\phi, \theta, \eta, \alpha) = \mathcal{L}_V(\phi) + \mathcal{L}_{\text{V-MPO}}(\theta, \eta, \alpha), \tag{10}$$

where $\phi$ are the parameters of the value network, $\theta$ the parameters of the policy network, and $\eta$ and $\alpha$ are Lagrange multipliers. In practice, the policy and value networks share most of their parameters in the form of a shared convolutional network (a ResNet) and recurrent LSTM core, and are optimized together (Fig. 1b) (Mnih et al., 2016). We note, however, that the value network parameters $\phi$ are considered fixed for the policy improvement loss, and gradients are not propagated.

The policy evaluation loss for the value function, $\mathcal{L}_V(\phi)$, is the standard regression to $n$-step returns and is given by Eq. 1 above. The policy improvement loss $\mathcal{L}_{\text{V-MPO}}(\theta, \eta, \alpha)$ is given by

$$\mathcal{L}_{\text{V-MPO}}(\theta, \eta, \alpha) = \mathcal{L}_\pi(\theta) + \mathcal{L}_\eta(\eta) + \mathcal{L}_\alpha(\theta, \alpha). \tag{11}$$

Here the policy loss is the weighted maximum likelihood loss

$$\mathcal{L}_\pi(\theta) = -\sum_{s,a\sim\tilde{\mathcal{D}}} \psi(s,a) \log \pi_\theta(a|s), \qquad \psi(s,a) = \frac{\exp\left(\frac{A^{\text{target}}(s,a)}{\eta}\right)}{\sum_{s,a\sim\tilde{\mathcal{D}}} \exp\left(\frac{A^{\text{target}}(s,a)}{\eta}\right)}, \tag{12}$$

where the advantages $A^{\text{target}}(s,a)$ for the target network policy $\pi_{\theta_{\text{target}}}(a|s)$ are estimated according to the standard method described above. The tilde over the dataset, $\tilde{\mathcal{D}}$, indicates that we take samples corresponding to the top half advantages in the batch of data. The $\eta$, or "temperature", loss is

$$\mathcal{L}_\eta(\eta) = \eta\epsilon_\eta + \eta \log\left[\frac{1}{|\tilde{\mathcal{D}}|} \sum_{s,a\sim\tilde{\mathcal{D}}} \exp\left(\frac{A^{\text{target}}(s,a)}{\eta}\right)\right]. \tag{13}$$

We perform the alternating optimization over $\theta$ and $\alpha$ while keeping a single loss function by alternately applying a "stop-gradient" to the Lagrange multiplier and KL term. Then the KL constraint, which can be viewed as a form of trust-region loss, is given by

$$\mathcal{L}_\alpha(\theta, \alpha) = \frac{1}{|\mathcal{D}|} \sum_{s\in\mathcal{D}} \left[\alpha\big(\epsilon_\alpha - \text{sg}\left[\left[D_{\text{KL}}\big(\pi_{\theta_{\text{target}}}(a|s)\|\pi_\theta(a|s)\big)\right]\right]\big) + \text{sg}[[\alpha]] D_{\text{KL}}\big(\pi_{\theta_{\text{target}}}(a|s)\|\pi_\theta(a|s)\big)\right], \tag{14}$$

where $\text{sg}[[\cdot]]$ indicates a stop gradient, i.e., that the enclosed term is assumed constant with respect to all variables. Note that here we use the full batch $\mathcal{D}$, not $\tilde{\mathcal{D}}$.

For continuous action spaces parametrized by Gaussian distributions, we use decoupled KL constraints for the M-step in Eq. 14 as in Abdolmaleki et al. (2018b); the precise form is given in Appendix C.

We used the Adam optimizer (Kingma & Ba, 2015) with default TensorFlow hyperparameters to optimize the total loss in Eq. 10. In particular, the learning rate was fixed at $10^{-4}$ for all experiments.

---

**Algorithm 1** V-MPO

    **given** Batch size $B$, unroll length $n$, $T_{\text{target}}$, KL bounds $\epsilon_\eta, \epsilon_\alpha$.
    **initialize** Network parameters $\theta_{\text{online}}, \phi_{\text{online}}$, Lagrange multipliers $\eta, \alpha$.
    **repeat**
        $\theta_{\text{target}} \leftarrow \theta_{\text{online}}$
        **for** $i = 1, \ldots, T_{\text{target}}$ **do**
            Use policy $\pi_{\theta_{\text{target}}}$ to act in the environment and collect $B$ trajectories $\tau$ of length $n$.
            Update $\theta_{\text{online}}, \phi_{\text{online}}, \eta, \alpha$ using Adam to minimize the total loss in Eq. 10.
            $\eta \leftarrow \max(\eta, \eta_{\min})$
            $\alpha \leftarrow \max(\alpha, \alpha_{\min})$
        **end for**
    **until** Fixed number of steps.

---

## 5 EXPERIMENTS

Details on the network architecture and hyperparameters used for each task are given in Appendix F.

### 5.1 DISCRETE ACTIONS: DMLAB, ATARI

*DMLab.* DMLab-30 (Beattie et al., 2016) is a collection of visually rich, partially observable 3D environments played from the first-person point of view. Like IMPALA, for DMLab we used pixel control as an auxiliary loss for representation learning (Jaderberg et al., 2017b; Hessel et al., 2018). However, we did not employ the optimistic asymmetric reward scaling used by previous IMPALA

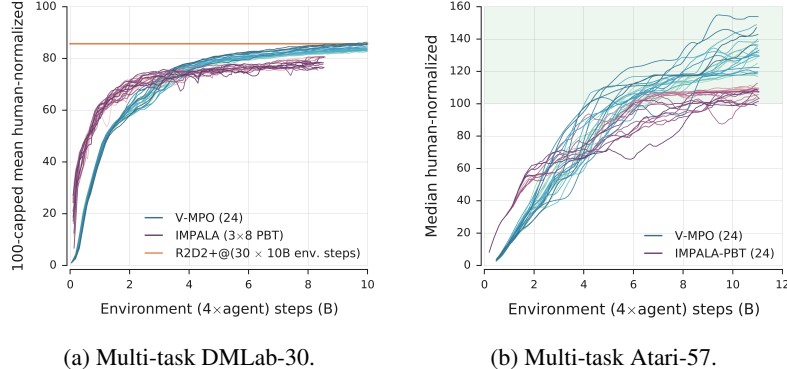

(a) Multi-task DMLab-30.    (b) Multi-task Atari-57.

Figure 2: (a) Multi-task DMLab-30. IMPALA results show 3 runs of 8 agents each; within a run hyperparameters were evolved via PBT. For V-MPO each line represents a set of hyperparameters that are fixed throughout training. The final result of R2D2+ trained for 10B environment steps on individual levels (Kapturowski et al., 2019) is also shown for comparison (orange line). (b) Multi-task Atari-57. In the IMPALA experiment, hyperparameters were evolved with PBT. For V-MPO each of the 24 lines represents a set of hyperparameters that were fixed throughout training, and all runs achieved a higher score than the best IMPALA run. Data for IMPALA ("Pixel-PopArt-IMPALA" for DMLab-30 and "PopArt-IMPALA" for Atari-57) was obtained from the authors of Hessel et al. (2018). Each agent step corresponds to 4 environment frames due to the action repeat.

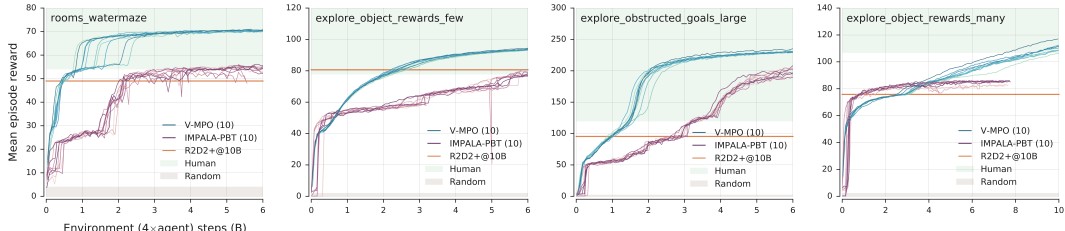

Figure 3: V-MPO trained on single example levels from DMLab-30, compared to IMPALA and more recent results from R2D2+, the larger, DMLab-specific version of R2D2 (Kapturowski et al., 2019). The IMPALA results include hyperparameter evolution with PBT.

experiments to aid exploration on a subset of the DMLab levels, by weighting positive rewards more than negative rewards (Espeholt et al., 2018; Hessel et al., 2018; Kapturowski et al., 2019). Unlike in Hessel et al. (2018) we also did not use population-based training (PBT) (Jaderberg et al., 2017a). Additional details for the settings used in DMLab can be found in Table 5 of the Appendix.

Fig. 2a shows the results for multi-task DMLab-30, comparing the V-MPO learning curves to data obtained from Hessel et al. (2018) for the PopArt IMPALA agent with pixel control. We note that the result for V-MPO at 10B environment frames across all levels matches the result for the Recurrent Replay Distributed DQN (R2D2) agent (Kapturowski et al., 2019) trained on *individual levels* for 10B environment steps per level. Fig. 3 shows example individual levels in DMLab where V-MPO achieves scores that are substantially higher than has previously been reported, for both R2D2 and IMPALA. The pixel-control IMPALA agents shown here were carefully tuned for DMLab and are similar to the "experts" used in Schmitt et al. (2018); in all cases these results match or exceed previously published results for IMPALA (Espeholt et al., 2018; Kapturowski et al., 2019).

*Atari.* The Atari Learning Environment (ALE) (Bellemare et al., 2012) is a collection of 57 Atari 2600 games that has served as an important benchmark for recent deep RL methods. We used the standard preprocessing scheme and a maximum episode length of 30 minutes (108,000 frames), see Table 6 in the Appendix. For the multi-task setting we followed Hessel et al. (2018) in setting the discount to zero on loss of life; for the example single tasks we did not employ this trick, since it

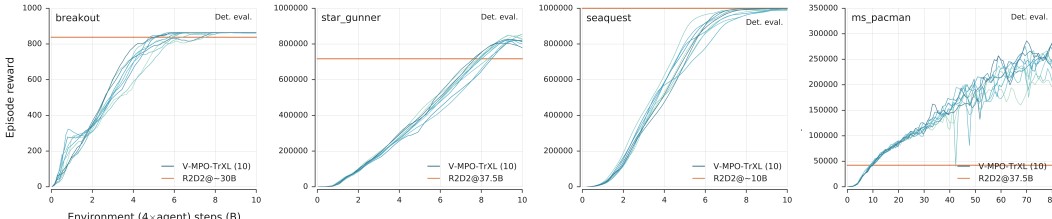

Figure 4: Example levels from Atari. In Breakout, V-MPO achieves the maximum score of 864 in every episode. No reward clipping was applied, and the maximum length of an episode was 30 minutes (108,000 frames). Supplementary video for Ms. Pacman: `https://bit.ly/2lWQBy5`

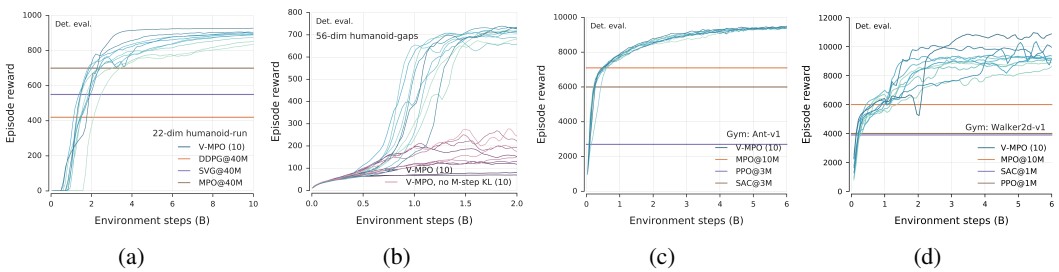

Figure 5: (a) Humanoid "run" from full state (Tassa et al., 2018) and (b) humanoid "gaps" from pixel observations (Merel et al., 2019). Purple curves are the same runs but without parametric KL constraints. Det. eval.: deterministic evaluation. Supplementary video for humanoid gaps: `https://bit.ly/2L9KZdS`. (c)-(d) Example OpenAI Gym tasks. See also Fig. 11 in the Appendix for Gym Humanoid-V1.

can prevent the agent from achieving the highest score possible by sacrificing lives. Similarly, while in the multi-task setting we followed previous work in clipping the maximum reward to 1.0, no such clipping was applied in the single-task setting in order to preserve the original reward structure. Additional details for the settings used in Atari can be found in Table 6 in the Appendix.

Fig. 2b shows the results for multi-task Atari-57, demonstrating that it is possible for a *single* agent to achieve "superhuman" median performance on Atari-57 in approximately 4 billion (~70 million per level) environment frames. Again, while we did not employ PBT in order to demonstrate that individual V-MPO runs can exceed the performance of a population of IMPALA agents, Fig. 6 shows that with population-based tuning of hyperparameters even higher performance is possible.

We also compare the performance of V-MPO on a few individual Atari levels to R2D2 (Kapturowski et al., 2019), which previously achieved some of the highest scores reported for Atari. Again, V-MPO can match or exceed previously reported scores while requiring fewer interactions with the environment. In Ms. Pacman, the final performance approaches 300,000 with a 30-minute timeout (and the maximum 1M without). Inspired by the argument in Kapturowski et al. (2019) that in a fully observable environment LSTMs enable the agent to utilize more useful representations than is available in the immediate observation, for the single-task setting we used a Transformer-XL (TrXL) (Dai et al., 2019) to replace the LSTM core. Unlike previous work for single Atari levels, we did not employ any reward clipping (Mnih et al., 2015; Espeholt et al., 2018) or nonlinear value function rescaling (Kapturowski et al., 2019).

## 5.2 CONTINUOUS CONTROL

To demonstrate V-MPO's effectiveness in high-dimensional, continuous action spaces, here we present examples of learning to control both a simulated humanoid with 22 degrees of freedom from full state observations and one with 56 degrees of freedom from pixel observations (Tassa et al., 2018; Merel et al., 2019). As shown in Fig. 5a, for the 22-dimensional humanoid V-MPO reliably achieves higher asymptotic returns than has previously been reported, including for Deep Deterministic Policy

Gradients (DDPG) (Lillicrap et al., 2015), Stochastic Value Gradients (SVG) (Heess et al., 2015), and MPO. These algorithms are far more sample-efficient but reach a lower final performance.

In the "gaps" task the 56-dimensional humanoid must run forward to match a target velocity of 4 m/s and jump over the gaps between platforms by learning to actuate joints with position-control (Merel et al., 2019). Previously, only an agent operating in the space of pre-learned motor primitives was able to solve the task from pixel observations (Merel et al., 2018; 2019); here we show that V-MPO can learn a challenging visuomotor task *from scratch* (Fig. 5b). For this task we also demonstrate the importance of the parametric KL constraint, without which the agent learns poorly.

In Figs. 5c-d we also show that V-MPO achieves the highest asymptotic performance reported for two OpenAI Gym tasks (Brockman et al., 2016). Again, MPO and Stochastic Actor-Critic (Haarnoja et al., 2018) are far more sample-efficient but reach a lower final performance.

These experiments are presented to demonstrate the existence of higher-return solutions than have previously been reported, and an algorithm, V-MPO, that can reliably converge to these solutions. However, in the future we desire algorithms that can do so while using fewer interactions with the environment.

## 6 Conclusion

In this work we have introduced a scalable on-policy deep reinforcement learning algorithm, V-MPO, that is applicable to both discrete and continuous control domains. For the results presented in this work neither importance weighting nor entropy regularization was used; moreover, since the size of neural network parameter updates is limited by KL constraints, we were also able to use the same learning rate for all experiments. This suggests that a scalable, performant RL algorithm may not require some of the tricks that have been developed over the past several years. Interestingly, both the original MPO algorithm for replay-based off-policy learning (Abdolmaleki et al., 2018a;b) and V-MPO for on-policy learning are derived from similar principles, providing evidence for the benefits of this approach as an alternative to popular policy gradient-based methods.

### Acknowledgments

We thank Lorenzo Blanco, Trevor Cai, Greg Wayne, Chloe Hillier, and Vicky Langston for their assistance and support.

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

## A    DERIVATION OF THE V-MPO TEMPERATURE LOSS

In this section we derive the E-step temperature loss in Eq. 22. To this end, we explicitly commit to the more specific improvement criterion in Eq. 5 by plugging into the original objective in Eq. 3. We seek $\psi(s, a)$ that minimizes

$$\mathcal{J}(\psi(s,a)) = D_{\text{KL}}\big(\psi(s,a)\|p_{\theta_{\text{old}}}(s,a|\mathcal{I}=1)\big) \tag{15}$$

$$\propto -\sum_{s,a} \psi(s,a) A^{\pi_{\theta_{\text{old}}}}(s,a) + \eta \sum_{s,a} \psi(s,a) \log \frac{\psi(s,a)}{p_{\theta_{\text{old}}}(s,a)} + \lambda \sum_{s,a} \psi(s,a) \tag{16}$$

where $\lambda = \eta \log p_{\theta_{\text{old}}}(\mathcal{I} = 1)$ after multiplying through by $\eta$, which up to this point in the derivation is given. We wish to automatically tune $\eta$ so as to enforce a bound $\epsilon_\eta$ on the KL term $D_{\text{KL}}\big(\psi(s,a)\|p_{\theta_{\text{old}}}(s,a)\big)$ multiplying it in Eq. 16, in which case the temperature optimization can also be viewed as a nonparametric trust region for the variational distribution with respect to the old distribution. We therefore consider the constrained optimization problem

$$\psi(s,a) = \arg \max_{\psi(s,a)} \sum_{s,a} \psi(s,a) A^{\pi_{\theta_{\text{old}}}}(s,a) \tag{17}$$

$$\text{s.t.} \sum_{s,a} \psi(s,a) \log \frac{\psi(s,a)}{p_{\theta_{\text{old}}}(s,a)} < \epsilon_\eta \text{ and } \sum_{s,a} \psi(s,a) = 1. \tag{18}$$

We can now use Lagrangian relaxation to transform the constrained optimization problem into one that maximizes the unconstrained objective

$$\mathcal{J}(\psi(s,a),\eta,\lambda) = \sum_{s,a} \psi(s,a) A^{\pi_{\theta_{\text{old}}}}(s,a) + \eta\left(\epsilon_\eta - \sum_{s,a}\psi(s,a)\log\frac{\psi(s,a)}{p_{\theta_{\text{old}}}(s,a)}\right) + \lambda\left(1 - \sum_{s,a}\psi(s,a)\right) \tag{19}$$

with $\eta \geq 0$. (Note we are re-using the variables $\eta$ and $\lambda$ for the new optimization problem.) Differentiating $\mathcal{J}$ with respect to $\psi(s,a)$ and setting equal to zero, we obtain

$$\psi(s,a) = p_{\theta_{\text{old}}}(s,a) \exp\left(\frac{A^{\pi_{\theta_{\text{old}}}}(s,a)}{\eta}\right) \exp\left(-1 - \frac{\lambda}{\eta}\right). \tag{20}$$

Normalizing over $s,a$ (using the freedom given by $\lambda$) then gives

$$\psi(s,a) = \frac{p_{\theta_{\text{old}}}(s,a) \exp\left(\frac{A^{\pi_{\theta_{\text{old}}}}(s,a)}{\eta}\right)}{\sum_{s,a} p_{\theta_{\text{old}}}(s,a) \exp\left(\frac{A^{\pi_{\theta_{\text{old}}}}(s,a)}{\eta}\right)}, \tag{21}$$

which reproduces the general solution Eq. 4 for our specific choice of policy improvement in Eq. 5. However, the value of $\eta$ can now be found by optimizing the corresponding dual function. Plugging Eq. 21 into the unconstrained objective in Eq. 19 gives rise to the $\eta$-dependent term

$$\mathcal{L}_\eta(\eta) = \eta\epsilon_\eta + \eta\log\left[\sum_{s,a} p_{\theta_{\text{old}}}(s,a)\exp\left(\frac{A^{\pi_{\theta_{\text{old}}}}(s,a)}{\eta}\right)\right]. \tag{22}$$

Replacing the expectation with samples from $p_{\theta_{\text{old}}}(s,a)$ in the batch of trajectories $\mathcal{D}$ leads to the loss in Eq. 13.

## B   M-STEP KL CONSTRAINT

Here we give a somewhat more formal motivation for the prior $\log p(\theta)$. Consider a normal prior $\mathcal{N}(\theta;\mu,\Sigma)$ with mean $\mu$ and covariance $\Sigma$. We choose $\Sigma^{-1} = \alpha F(\theta_{\text{old}})$ where $\alpha$ is a scaling parameter and $F(\theta_{\text{old}})$ is the Fisher information for $\pi_{\theta'}(a|s)$ evaluated at $\theta' = \theta_{\text{old}}$. Then $\log p(\theta) \approx -\alpha \times \frac{1}{2}(\theta - \theta_{\text{old}})^T F(\theta_{\text{old}})(\theta - \theta_{\text{old}}) + \{\text{term independent of } \theta\}$, where the first term is precisely the second-order approximation to the KL divergence $D_{\text{KL}}(\theta_{\text{old}}\|\theta)$. We now follow TRPO (Schulman et al., 2015) in heuristically approximating this as the state-averaged expression, $\mathbb{E}_{s\sim p(s)}\big[D_{\text{KL}}\big(\pi_{\theta_{\text{old}}}(a|s)\|\pi_\theta(a|s)\big)\big]$. We note that the KL divergence in either direction has the same second-order expansion, so our choice of KL is an empirical one (Abdolmaleki et al., 2018a).

## C   DECOUPLED KL CONSTRAINTS FOR CONTINUOUS CONTROL

As in Abdolmaleki et al. (2018b), for continuous action spaces parametrized by Gaussian distributions we use decoupled KL constraints for the M-step. This uses the fact that the KL divergence between two $d$-dimensional multivariate normal distributions with means $\mu_1, \mu_2$ and covariances $\Sigma_1, \Sigma_2$ can be written as

$$D_{\text{KL}}\big(\mathcal{N}(\mu_1,\Sigma_1)\|\mathcal{N}(\mu_2,\Sigma_2)\big) = \frac{1}{2}\left[(\mu_2-\mu_1)^T\Sigma_1^{-1}(\mu_2-\mu_1)+\text{Tr}(\Sigma_2^{-1}\Sigma_1)-d+\log\frac{|\Sigma_2|}{|\Sigma_1|}\right], \tag{23}$$

where $|\cdot|$ is the matrix determinant. Since the first distribution and hence $\Sigma_1$ in the KL divergence of Eq. 9 depends on the old target network parameters, we see that we can separate the overall KL divergence into a mean component and a covariance component:

$$D_{\text{KL}}^{\mu}\left(\pi_{\theta_{\text{old}}}\|\pi_{\theta}\right) = \frac{1}{2}(\mu_{\theta} - \mu_{\theta_{\text{old}}})^T\Sigma_{\theta_{\text{old}}}^{-1}(\mu_{\theta} - \mu_{\theta_{\text{old}}}), \tag{24}$$

$$D_{\text{KL}}^{\Sigma}\left(\pi_{\theta_{\text{old}}}\|\pi_{\theta}\right) = \frac{1}{2}\left[\text{Tr}(\Sigma_{\theta}^{-1}\Sigma_{\theta_{\text{old}}}) - d + \log\frac{|\Sigma_{\theta}|}{|\Sigma_{\theta_{\text{old}}}|}\right]. \tag{25}$$

With the replacement $D_{\text{KL}}\left(\pi_{\theta_{\text{old}}}\|\pi_{\theta}\right) \rightarrow D_{\text{KL}}^{C}\left(\pi_{\theta_{\text{old}}}\|\pi_{\theta}\right)$ for $C = \mu, \Sigma$ and corresponding $\alpha \rightarrow \alpha_{\mu}, \alpha_{\Sigma}$, we obtain the total loss

$$\mathcal{L}_{\text{V-MPO}}(\theta, \eta, \alpha_{\mu}, \alpha_{\Sigma}) = \mathcal{L}_{\pi}(\theta) + \mathcal{L}_{\eta}(\eta) + \mathcal{L}_{\alpha_{\mu}}(\theta, \alpha_{\mu}) + \mathcal{L}_{\alpha_{\Sigma}}(\theta, \alpha_{\Sigma}), \tag{26}$$

where $\mathcal{L}_{\pi}(\theta)$ and $\mathcal{L}_{\eta}(\eta)$ are the same as before. Note, however, that unlike in Abdolmaleki et al. (2018a) we do not decouple the policy loss.

We generally set $\epsilon_{\Sigma}$ to be much smaller than $\epsilon_{\mu}$ (see Table 7). Intuitively, this allows the policy to learn quickly in action space while preventing premature collapse of the policy, and, conversely, increasing "exploration" without moving in action space.

## D  RELATION TO SUPERVISED POLICY UPDATE

Like V-MPO, Supervised Policy Update (SPU) (Vuong et al., 2019) adopts the strategy of first solving a nonparametric constrained optimization problem exactly, then fitting a neural network to the resulting solution via a supervised loss function. There is, however, an important difference from V-MPO, which we describe here.

In SPU, the KL loss, which is the *sole* loss in SPU, leads to a parametric optimization problem that is equivalent to the nonparametric optimization problem posed initially. To see this, we observe that the SPU loss seeks parameters (note the direction of the KL divergence)

$$\theta^* = \arg\min_{\theta}\sum_s d^{\pi_{\theta_k}}(s)D_{\text{KL}}\left(\pi_{\theta}(a|s)\|\pi^{\lambda}(a|s)\right) \tag{27}$$

$$= \arg\min_{\theta}\sum_s d^{\pi_{\theta_k}}(s)\sum_a \pi_{\theta}(a|s)\log\left[\frac{\pi_{\theta}(a|s)}{\pi_{\theta_k}(a|s)\exp\left(A^{\pi_{\theta_k}}(s,a)/\lambda\right)/Z_{\lambda}(s)}\right] \tag{28}$$

$$= \arg\min_{\theta}\sum_s d^{\pi_{\theta_k}}(s)\sum_a \left[\pi_{\theta}(a|s)\log\frac{\pi_{\theta}(a|s)}{\pi_{\theta_k}(a|s)} - \frac{1}{\lambda}\pi_{\theta}(a|s)A^{\pi_{\theta_k}}(s,a)\right] + \{\text{ constant terms }\}. \tag{29}$$

Multiplying by $\lambda$ since it can be treated as a constant up to this point, we then see that this corresponds exactly to the (Lagrangian form) of the problem

$$\theta^* = \arg\max_{\theta}\sum_s d^{\pi_{\theta_k}}(s)\sum_a \pi_{\theta}(a|s)A^{\pi_{\theta_k}}(s,a) \tag{30}$$

$$\text{s.t.}\sum_s d^{\pi_{\theta_k}}(s)D_{\text{KL}}\left(\pi_{\theta}(a|s)\|\pi_{\theta_k}(a|s)\right) < \epsilon, \tag{31}$$

which is the original nonparametric problem posed in Vuong et al. (2019).

## E  IMPORTANCE-WEIGHTING FOR OFF-POLICY CORRECTIONS

The network that generates the data may lag behind the target network in common distributed, asynchronous implementations (Espeholt et al., 2018). We can compensate for this by multiplying the exponentiated advantages by importance weights $\rho(s, a)$:

$$\psi(s, a) = \frac{\rho(s, a)p_{\theta_{\mathcal{D}}}(s, a)\exp\left(\frac{A^{\pi_{\theta_{\mathcal{D}}}}(s,a)}{\eta}\right)}{\sum_{s,a}\rho(s, a)p_{\theta_{\mathcal{D}}}(s, a)\exp\left(\frac{A^{\pi_{\theta_{\mathcal{D}}}}(s,a)}{\eta}\right)}, \tag{32}$$

$$\mathcal{L}_{\eta}(\eta) = \eta\epsilon_{\eta} + \eta\log\left[\sum_{s,a}\rho(s, a)p_{\theta_{\mathcal{D}}}(s, a)\exp\left(\frac{A^{\pi_{\theta_{\mathcal{D}}}}(s,a)}{\eta}\right)\right], \tag{33}$$

where $\theta_{\mathcal{D}}$ are the parameters of the behavior policy that generated $\mathcal{D}$ and which may be different from $\theta_{\text{target}}$. The clipped importance weights $\rho(s, a)$ are given by

$$\rho(s, a) = \min\left(1, \frac{\pi_{\theta_{\text{old}}}(a|s)}{\pi_{\theta_{\mathcal{D}}}(a|s)}\right). \tag{34}$$

As was the case with V-trace for the value function, we did not find it necessary to use importance weighting and all experiments presented in this work did not use them for the sake of simplicity.

## F   NETWORK ARCHITECTURE AND HYPERPARAMETERS

For DMLab the visual observations were 72×96 RGB images, while for Atari the observations were 4 stacked frames of 84×84 grayscale images. The ResNet used to process visual observations is similar to the 3-section ResNet used in Hessel et al. (2018), except the number of channels was multiplied by 4 in each section, so that the number of channels were (64, 128, 128) (Schmitt et al., 2019). For individual DMLab levels we used the same number of channels as Hessel et al. (2018), i.e., (16, 32, 32). Each section consisted of a convolution and $3 \times 3$ max-pooling operation (stride 2), followed by residual blocks of size 2, i.e., a convolution followed by a ReLU nonlinearity, repeated twice, and a skip connection from the input residual block input to the output. The entire stack was passed through one more ReLU nonlinearity. All convolutions had a kernel size of 3 and a stride of 1. For the humanoid control tasks from vision, the number of channels in each section were (16, 32, 32).

Since some of the levels in DMLab require simple language processing, for DMLab the agents contained an additional 256-unit LSTM receiving an embedding of hashed words as input. The output of the language LSTM was then concatenated with the output of the visual processing pathway as well as the previous reward and action, then fed to the main LSTM.

For multi-task DMLab we used a 3-layer LSTM, each with 256 units, and an unroll length of 95 with batch size 128. For the single-task setting we used a 2-layer LSTM. For multi-task Atari and the 56-dimensional humanoid-gaps control task a single 256-unit LSTM was used, while for the 22-dimensional humanoid-run task the core consisted only of a 2-layer MLP with 512 and 256 units (no LSTM). For single-task Atari a Transformer-XL was used in place of the LSTM. Note that we followed Radford et al. (2019) in placing the layer normalization on only the inputs to each sub-block. For Atari the unroll length was 63 with a batch size of 128. For both humanoid control tasks the batch size was 64, but the unroll length was 40 for the 22-dimensional humanoid and 63 for the 56-dimensional humanoid.

In all cases the policy logits (for discrete actions) and Gaussian distribution parameters (for continuous actions) consisted of a 256-unit MLP followed by a linear readout, and similarly for the value function. For discrete actions we initialized the linear policy layer with zero weights and biases to ensure a uniform policy at the start of training.

The initial values for the Lagrange multipliers in the V-MPO loss are given in Table 1

*Implementation note.* We implemented V-MPO in an actor-learner framework (Espeholt et al., 2018) that utilizes TF-Replicator (Buchlovsky et al., 2019) for distributed training on TPU 8-core and 16-core configurations (Google, 2018). One practical consequence of this is that a full batch of data $\mathcal{D}$ was in fact split into 8 or 16 minibatches, one per core/replica, and the overall result obtained by averaging the computations performed for each minibatch. More specifically, the determination of the highest advantages and the normalization of the nonparametric distribution, Eq. 12, is performed within minibatches. While it is possible to perform the full-batch computation by utilizing cross-replica communication, we found this to be unnecessary.

*DMLab action set.* Ignoring the "jump" and "crouch" actions which we do not use, an action in the native DMLab action space consists of 5 integers whose meaning and allowed values are given in Table 2. Following previous work on DMLab (Hessel et al., 2018), we used the reduced action set given in Table 3 with an action repeat of 4.

| HYPERPARAMETER | VALUE | | |
|---|---|---|---|
| | DMLab | Atari | Continuous control |
| Initial $\eta$ | 1.0 | 1.0 | 1.0 |
| Initial $\alpha$ | 5.0 | 5.0 | - |
| Initial $\alpha_\mu$ | - | - | 1.0 |
| Initial $\alpha_\Sigma$ | - | - | 1.0 |

Table 1: Values for common V-MPO parameters.

| ACTION NAME | RANGE |
|---|---|
| LOOK_LEFT_RIGHT_PIXELS_PER_FRAME | [-512, 512] |
| LOOK_DOWN_UP_PIXELS_PER_FRAME | [-512, 512] |
| STRAFE_LEFT_RIGHT | [-1, 1] |
| MOVE_BACK_FORWARD | [-1, 1] |
| FIRE | [0, 1] |

Table 2: Native action space for DMLab. See `https://github.com/deepmind/lab/blob/master/docs/users/actions.md` for more details.

| ACTION | NATIVE DMLAB ACTION |
|---|---|
| Forward (FW) | [ 0, 0, 0, 1, 0] |
| Backward (BW) | [ 0, 0, 0, -1, 0] |
| Strafe left | [ 0, 0, -1, 0, 0] |
| Strafe right | [ 0, 0, 1, 0, 0] |
| Small look left (LL) | [-10, 0, 0, 0, 0] |
| Small look right (LR) | [ 10, 0, 0, 0, 0] |
| Large look left (LL ) | [-60, 0, 0, 0, 0] |
| Large look right (LR) | [ 60, 0, 0, 0, 0] |
| Look down | [ 0, 10, 0, 0, 0] |
| Look up | [ 0, -10, 0, 0, 0] |
| FW + small LL | [-10, 0, 0, 1, 0] |
| FW + small LR | [ 10, 0, 0, 1, 0] |
| FW + large LL | [-60, 0, 0, 1, 0] |
| FW + large LR | [ 60, 0, 0, 1, 0] |
| Fire | [ 0, 0, 0, 0, 1] |

Table 3: Reduced action set for DMLab from Hessel et al. (2018).

| LEVEL NAME | EPISODE REWARD | | HUMAN-NORMALIZED | |
| | IMPALA | V-MPO | IMPALA | V-MPO |
|---|---|---|---|---|
| alien | 1163.00 ± 148.43 | 2332.00 ± 290.16 | 13.55 ± 2.15 | 30.50 ± 4.21 |
| amidar | 192.50 ± 9.16 | 423.60 ± 20.53 | 10.89 ± 0.53 | 24.38 ± 1.20 |
| assault | 4215.30 ± 294.51 | 1225.90 ± 60.64 | 768.46 ± 56.68 | 193.13 ± 11.67 |
| asterix | 4180.00 ± 303.91 | 9955.00 ± 2043.48 | 47.87 ± 3.66 | 117.50 ± 24.64 |
| asteroids | 3473.00 ± 381.30 | 2982.00 ± 164.35 | 5.90 ± 0.82 | 4.85 ± 0.35 |
| atlantis | 997530.00 ± 3552.89 | 940310.00 ± 6085.96 | 6086.50 ± 21.96 | 5732.81 ± 37.62 |
| bank_heist | 1329.00 ± 2.21 | 1563.00 ± 15.81 | 177.94 ± 0.30 | 209.61 ± 2.14 |
| battle_zone | 43900.00 ± 4738.04 | 61400.00 ± 5958.52 | 119.27 ± 13.60 | 169.52 ± 17.11 |
| beam_rider | 4598.00 ± 618.09 | 3868.20 ± 666.55 | 25.56 ± 3.73 | 21.16 ± 4.02 |
| berzerk | 1018.00 ± 72.63 | 1424.00 ± 150.93 | 35.68 ± 2.90 | 51.87 ± 6.02 |
| bowling | 63.60 ± 0.84 | 27.60 ± 0.62 | 29.43 ± 0.61 | 3.27 ± 0.45 |
| boxing | 93.10 ± 0.94 | 100.00 ± 0.00 | 775.00 ± 7.86 | 832.50 ± 0.00 |
| breakout | 484.30 ± 57.24 | 400.70 ± 18.82 | 1675.69 ± 198.77 | 1385.42 ± 65.36 |
| centipede | 6037.90 ± 994.99 | 3015.00 ± 404.97 | 39.76 ± 10.02 | 9.31 ± 4.08 |
| chopper_command | 4250.00 ± 417.91 | 4340.00 ± 714.45 | 52.29 ± 6.35 | 53.66 ± 10.86 |
| crazy_climber | 100440.00 ± 9421.56 | 116760.00 ± 5312.12 | 357.94 ± 37.61 | 423.09 ± 21.21 |
| defender | 41585.00 ± 4194.42 | 98395.00 ± 17552.17 | 244.78 ± 26.52 | 604.01 ± 110.99 |
| demon_attack | 77880.00 ± 8798.44 | 20243.00 ± 5434.41 | 4273.35 ± 483.72 | 1104.56 ± 298.77 |
| double_dunk | -0.80 ± 0.31 | 12.60 ± 1.94 | 809.09 ± 14.08 | 1418.18 ± 88.19 |
| enduro | 1187.90 ± 76.10 | 1453.80 ± 104.37 | 138.05 ± 8.84 | 168.95 ± 12.13 |
| fishing_derby | 21.60 ± 3.46 | 33.80 ± 2.10 | 213.77 ± 6.54 | 236.79 ± 3.96 |
| freeway | 32.10 ± 0.17 | 33.20 ± 0.28 | 108.45 ± 0.58 | 112.16 ± 0.93 |
| frostbite | 250.00 ± 0.00 | 260.00 ± 0.00 | 4.33 ± 0.00 | 4.56 ± 0.00 |
| gopher | 11720.00 ± 1687.71 | 7576.00 ± 973.13 | 531.92 ± 78.32 | 339.62 ± 45.16 |
| gravitar | 1095.00 ± 232.75 | 3125.00 ± 191.87 | 29.01 ± 7.32 | 92.88 ± 6.04 |
| hero | 13159.50 ± 68.90 | 29196.50 ± 752.06 | 40.71 ± 0.23 | 94.53 ± 2.52 |
| ice_hockey | 4.80 ± 1.31 | 10.60 ± 2.00 | 132.23 ± 10.83 | 180.17 ± 16.50 |
| jamesbond | 1015.00 ± 91.39 | 3805.00 ± 595.92 | 360.12 ± 33.38 | 1379.11 ± 217.65 |
| kangaroo | 1780.00 ± 18.97 | 12790.00 ± 629.52 | 57.93 ± 0.64 | 427.02 ± 21.10 |
| krull | 9738.00 ± 360.95 | 7359.00 ± 1064.84 | 762.53 ± 33.81 | 539.67 ± 99.75 |
| kung_fu_master | 44340.00 ± 2898.70 | 38620.00 ± 2346.48 | 196.11 ± 12.90 | 170.66 ± 10.44 |
| montezuma_revenge | 0.00 ± 0.00 | 0.00 ± 0.00 | 0.00 ± 0.00 | 0.00 ± 0.00 |
| ms_pacman | 1953.00 ± 227.12 | 2856.00 ± 324.54 | 24.77 ± 3.42 | 38.36 ± 4.88 |
| name_this_game | 5708.00 ± 354.92 | 9295.00 ± 679.83 | 59.33 ± 6.17 | 121.64 ± 11.81 |
| phoenix | 37030.00 ± 6415.95 | 19560.00 ± 1843.44 | 559.60 ± 98.99 | 290.05 ± 28.44 |
| pitfall | -4.90 ± 2.34 | -2.80 ± 1.40 | 3.35 ± 0.04 | 3.39 ± 0.02 |
| pong | 20.80 ± 0.19 | 21.00 ± 0.00 | 117.56 ± 0.54 | 118.13 ± 0.00 |
| private_eye | 100.00 ± 0.00 | 100.00 ± 0.00 | 0.11 ± 0.00 | 0.11 ± 0.00 |
| qbert | 5512.50 ± 741.08 | 15297.50 ± 1244.47 | 40.24 ± 5.58 | 113.86 ± 9.36 |
| riverraid | 8237.00 ± 97.09 | 11160.00 ± 733.06 | 43.72 ± 0.62 | 62.24 ± 4.65 |
| road_runner | 28440.00 ± 1215.99 | 51060.00 ± 1560.72 | 362.91 ± 15.52 | 651.67 ± 19.92 |
| robotank | 29.60 ± 2.15 | 46.80 ± 3.42 | 282.47 ± 22.22 | 459.79 ± 35.29 |
| seaquest | 1888.00 ± 63.26 | 9953.00 ± 973.02 | 4.33 ± 0.15 | 23.54 ± 2.32 |
| skiing | -16244.00 ± 592.28 | -15438.10 ± 1573.39 | 6.69 ± 4.64 | 13.01 ± 12.33 |
| solaris | 1794.00 ± 279.04 | 2194.00 ± 417.91 | 5.03 ± 2.52 | 8.64 ± 3.77 |
| space_invaders | 793.50 ± 90.61 | 1771.50 ± 201.95 | 42.45 ± 5.96 | 106.76 ± 13.28 |
| star_gunner | 44860.00 ± 5157.74 | 60120.00 ± 1953.60 | 461.05 ± 53.80 | 620.24 ± 20.38 |
| surround | 2.50 ± 1.04 | 4.00 ± 0.62 | 75.76 ± 6.31 | 84.85 ± 3.74 |
| tennis | -0.10 ± 0.09 | 23.10 ± 0.26 | 152.90 ± 0.61 | 302.58 ± 1.69 |
| time_pilot | 10890.00 ± 787.46 | 22330.00 ± 2443.11 | 440.77 ± 47.40 | 1129.42 ± 147.07 |
| tutankham | 218.50 ± 13.53 | 254.60 ± 9.99 | 132.59 ± 8.66 | 155.70 ± 6.40 |
| up_n_down | 175083.00 ± 16341.05 | 82913.00 ± 12142.08 | 1564.09 ± 146.43 | 738.18 ± 108.80 |
| venture | 0.00 ± 0.00 | 0.00 ± 0.00 | 0.00 ± 0.00 | 0.00 ± 0.00 |
| video_pinball | 59898.40 ± 23875.14 | 198845.20 ± 98768.54 | 339.02 ± 135.13 | 1125.46 ± 559.03 |
| wizard_of_wor | 6960.00 ± 1730.97 | 7890.00 ± 1595.77 | 152.55 ± 41.28 | 174.73 ± 38.06 |
| yars_revenge | 12825.70 ± 2065.90 | 41271.70 ± 4726.72 | 18.90 ± 4.01 | 74.16 ± 9.18 |
| zaxxon | 11520.00 ± 646.81 | 18820.00 ± 754.69 | 125.67 ± 7.08 | 205.53 ± 8.26 |
| Median | | | 117.56 | 155.70 |

Table 4: Multi-task Atari-57 scores by level after 11.4B total (200M per level) environment frames. All entries show mean ± standard deviation. Data for IMPALA ("PopArt-IMPALA") was obtained from the authors of Hessel et al. (2018). Human-normalized scores are calculated as $(E-R)/(H-R)\times100$, where $E$ is the episode reward, $R$ the episode reward obtained by a random agent, and $H$ is the episode reward obtained by a human.

| SETTING | SINGLE-TASK | MULTI-TASK |
|---|:---:|:---:|
| Agent discount | 0.99 | |
| Image height | 72 | |
| Image width | 96 | |
| Number of action repeats | 4 | |
| Number of LSTM layers | 2 | 3 |
| Pixel-control cost | $2 \times 10^{-3}$ | |
| $T_{\text{target}}$ | 10 | |
| $\epsilon_\eta$ | 0.1 | 0.5 |
| $\epsilon_\alpha$ (log-uniform) | $[0.001,\ 0.01)$ | $[0.01,\ 0.1)$ |

Table 5: Settings for DMLab.

| SETTING | SINGLE-TASK | MULTI-TASK |
|---|:---:|:---:|
| Environment discount on end of life | 1 | 0 |
| Agent discount | 0.997 | 0.99 |
| Clipped reward range | no clipping | $[-1, 1]$ |
| Max episode length | 30 mins (108,000 frames) | |
| Image height | 84 | |
| Image width | 84 | |
| Grayscale | True | |
| Number of stacked frames | 4 | |
| Number of action repeats | 4 | |
| TrXL: Key/Value size | 32 | . |
| TrXL: Number of heads | 8 | . |
| TrXL: Number of layers | 8 | . |
| TrXL: MLP size | 512 | . |
| $T_{\text{target}}$ | 1000 | 100 |
| $\epsilon_\eta$ | $1 \times 10^{-1}$ | |
| $\epsilon_\alpha$ (log-uniform) | $[0.005, 0.01)$ | $[0.001, 0.01)$ |

Table 6: Settings for Atari. TrXL: Transformer-XL.

| SETTING | HUMANOID-PIXELS | HUMANOID-STATE | OPENAI GYM |
|---|:---:|:---:|:---:|
| Agent discount | | 0.99 | |
| Unroll length | 63 | 63 | 39 |
| Image height | 64 | . | . |
| Image width | 64 | . | . |
| Target update period | | 100 | |
| $\epsilon_\eta$ | 0.1 | | 0.01 |
| $\epsilon_{\alpha_\mu}$ (log-uniform) | $[0.01,\ 1.0)$ | $[0.05,\ 0.5]$ | $[0.005,\ 0.01]$ |
| $\epsilon_{\alpha_\Sigma}$ (log-uniform) | $[5 \times 10^{-6},\ 5 \times 10^{-5})$ | $[10^{-5},\ 5 \times 10^{-5})$ | $[5 \times 10^{-6},\ 5 \times 10^{-5})$ |

Table 7: Settings for continuous control. For the humanoid gaps task from pixels the physics time step was 5 ms and the control time step 30 ms.

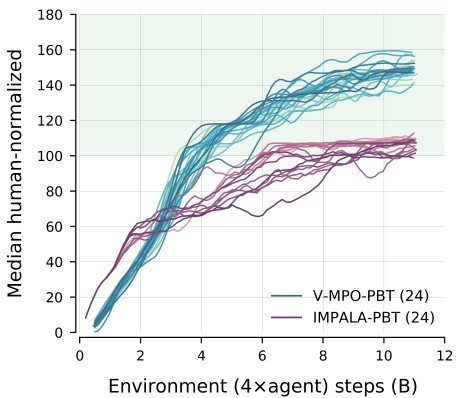

Figure 6: Multi-task Atari-57 with population-based training (PBT) (Jaderberg et al., 2017a). All settings of the PBT experiment were the same as without except the learning rates were also sampled log-uniformly from $[8 \times 10^{-5},\ 3 \times 10^{-4})$ and $\epsilon_\eta$ from $[0.05,\ 0.5)$. Along with $\epsilon_\alpha$ sampled log-uniformly from $[0.001,\ 0.01)$ as in the original experiment, hyperparameters were evolved via copy and mutation operators roughly once every $4 \times 10^8$ environment frames.

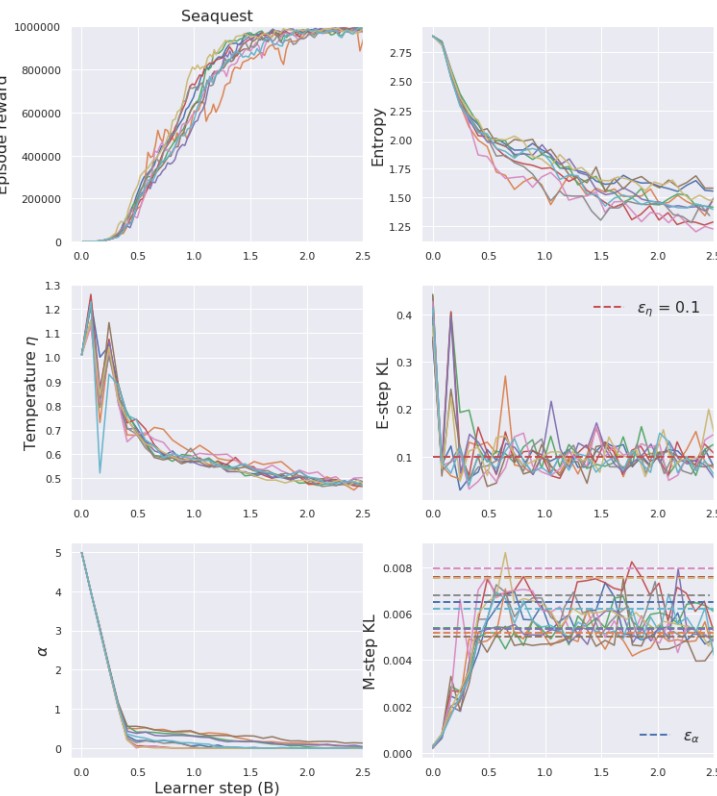

Figure 7: KL constraints during optimization for the Seaquest example in Fig. 4c. Values are subsampled but not smoothed to show the variability.

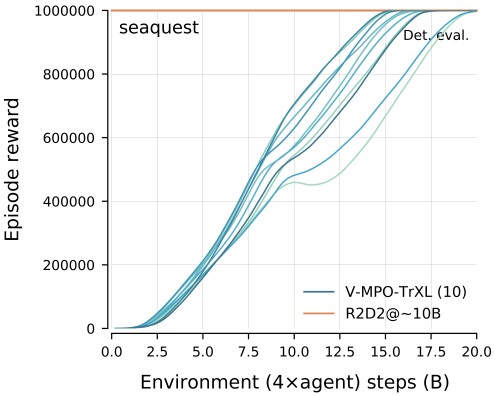

Figure 8: Same as Fig. 4c (Atari Seaquest), but trained with uniform weights on the top 50% of advantages.

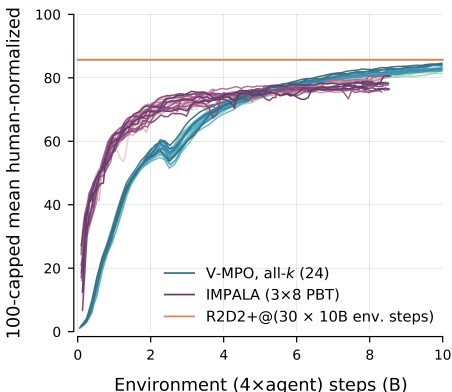

Figure 9: Same as Fig. 2a (multi-task DMLab-30), but trained without top-$k$, i.e., all advantages are used in the E-step. Note the small dip in the middle is due to a pause in the experiment and resetting of the human-normalized scores.

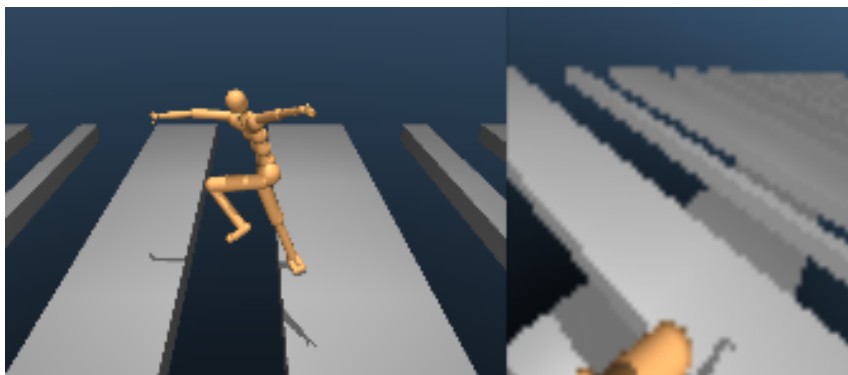

Figure 10: Example frame from the humanoid gaps task, with the agent's 64×64 first-person view on the right. The proprioceptive information provided to the agent in addition to the primary pixel observation consisted of joint angles and velocities, root-to-end-effector vectors, root-frame velocity, rotational velocity, root-frame acceleration, and the 3D orientation relative to the $z$-axis.

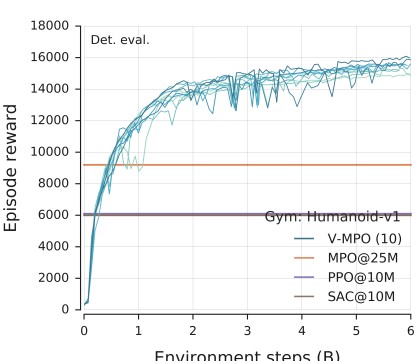

Figure 11: 17-dimensional Humanoid-V1 task in OpenAI Gym.

