# OpenReview forum: "V-MPO: On-Policy Maximum a Posteriori Policy Optimization for Discrete and Continuous Control"
_ICLR.cc/2020/Conference — Accept (Poster)_

### Official Review · AnonReviewer3 · 2019-10-22
**Official Blind Review #3**

**Rating:** 6

**Review:**

Summary: This paper presents the V-MPO algorithm for on-policy reinforcement learning that can handle both continuous/discrete control, single/multi-task learning and use both low dimensional states and pixels.

V-MPO adapts MPO, a recent off-policy deep reinforcement learning algorithm, to the on-policy setting with the following changes: (1) In policy evaluation step, instead of using state-action value Q estimated from off-policy replay data, the authors use on-policy data to estimate the state value V; (2) In E-step of policy learning, construct the target distribution q(a|s) with estimated advantages; and (3) in M-step, rewrite the KL divergence constraints with Lagrangian relaxation and alternate between optimising policy and the multiplier alpha.

The experiments are two-fold: (1) For discrete control, a multi-task problem setting is considered (DMLab-30 and Atari-57). V-MPO shows improved performance compared to IMPALA and R2D2, in terms of asymptotic scores. The result also suggest good stability for different hyper-parameters. (2) In continuous problems, the experiments follow the single-task problem setting, including Humanoids tasks and OpenAI Gym tasks. V-MPO achieves higher asymptotic returns compared with standard Deep RL algorithms (MPO, SAC, PPO). However, it has a lower sample-efficiency, especially compared to off-policy algorithms (MPO and SAC).

Pros:
+ This paper demonstrates successful adaption of MPO to the on-policy problem setting and achieves better asymptotic score comparing to baseline methods.
++ Results are achieved in a relatively hyper-parameter insensitive manner. And some commonly used tricks like entropy regularisation are not necessary.
+ Along with the previous results of MPO, this demonstrates an alternative framework of policy gradient-based RL: first construct a nonparametric target behavioural distribution and then move the parametric policy towards this distribution.

Cons and Questions:
1. The comparison between V-MPO with other single-task deep RL methods is non-standard because: (1) V-MPO is trained with far more samples, and (2) only asymptotic score is reported for baselines. It would be more informative to add the learning curve of baselines methods to show sample-efficiency and convergence properties. One may wonder if the baselines would provide better performance than reported if provided with a comparable number of steps in an appropriate way (E.g., seed search, hyper parameter sweeps).
2. The proposed adaption of MPO from off-policy to on-policy applies to both discrete and continuous problem settings. However, the designed experiments include only the discrete multi-task setting and continuous single-task setting. How does V-MPO compare to other methods in single-task discrete problems? This may enable comparison to a wider set of prior work.
3. Only top 50% advantages samples are used for generating target distribution in E-step, and all samples are used for policy updates in M-step. Does this mismatch between E-step and M-step sample pool make a difference in optimisation?
4. Some small tricks we used like PopArt, Top-K advantage. It’s not clear if these are things that (were/could-also-be used) in competitor methods, and how important these are to achieve the good results shown here. At least an ablation study on their impact for V-MPO would be nice.

Minor:
- The mixture use of subscripts “old” and “target” is rather confusing, for example, in equation (5) and (15).

**Experience Assessment:**

I have published one or two papers in this area.

**Review Assessment: Checking Correctness Of Derivations And Theory:**

I assessed the sensibility of the derivations and theory.

**Review Assessment: Checking Correctness Of Experiments:**

I assessed the sensibility of the experiments.

**Review Assessment: Thoroughness In Paper Reading:**

I read the paper at least twice and used my best judgement in assessing the paper.

---

> ### Author Response · Authors · 2019-11-13
> **Response to Reviewer #3 (1/2)**
>
> We thank the reviewer for their thoughtful questions and comments, and we appreciate their thorough understanding of the paper.
>
> “The comparison between V-MPO with other single-task deep RL methods is non-standard because: (1) V-MPO is trained with far more samples, and (2) only asymptotic score is reported for baselines. It would be more informative to add the learning curve of baselines methods to show sample-efficiency and convergence properties. One may wonder if the baselines would provide better performance than reported if provided with a comparable number of steps in an appropriate way (E.g., seed search, hyper parameter sweeps).”
>
> We have wondered this ourselves; however, it is difficult to do justice to other algorithms (see, for example, the discussion in https://openreview.net/forum?id=H1gdF34FvS), including all of the tricks used by other works, so we have made the baseline numbers explicit and encourage others to show that these numbers could be improved if optimized for the high-data regime.
>
> On the other hand, we acknowledge that this version of the algorithm is likely to be  outperformed by other methods in the extremely low data regime. Even in DMLab-30 and Atari-57, it is evident that with the current set of hyperparameters V-MPO underperforms IMPALA early on but eventually reaches substantially higher scores later.
>
> Our goal was to show that these long-studied tasks have solutions with far higher returns than has previously been reported, and to demonstrate the existence of an algorithm, namely V-MPO, that can reliably reach them. We think this is interesting to know. It is unclear to us whether other algorithms, if run for longer, could find these solutions - we hope others will try!
>
> “However, the designed experiments include only the discrete multi-task setting and continuous single-task setting. How does V-MPO compare to other methods in single-task discrete problems? This may enable comparison to a wider set of prior work.”
>
> Precisely for this reason, Fig. 3 (in the updated numbering) shows V-MPO trained on example *single* tasks in DMLab, and Fig. 4 shows V-MPO applied to example *single* tasks in Atari. In hindsight this is not so clear in the caption and we have changed this in the updated paper.
>
> “Only top 50% advantages samples are used for generating target distribution in E-step, and all samples are used for policy updates in M-step. Does this mismatch between E-step and M-step sample pool make a difference in optimisation?”
>
> To clarify, only the samples corresponding to the top 50% of advantages are also used for the weighted maximum likelihood fitting in the M-step, as these are the weights generated by the E-step and the bottom half of samples get zero weight. The KL prior in the M-step does use all samples, but this is not a problem as this parametric prior is separate from the nonparametric target distribution proposed in the E-step.

---

> > ### Author Response · Authors · 2019-11-13
> > **Response to Reviewer #3 (2/2)**
> >
> > “Some small tricks we used like PopArt, Top-K advantage. It’s not clear if these are things that (were/could-also-be used) in competitor methods, and how important these are to achieve the good results shown here. At least an ablation study on their impact for V-MPO would be nice.”
> >
> > We acknowledge that V-MPO benefited from both PopArt normalization and top-k, otherwise we would not have kept them! The benefits vary across task; as shown in Fig. 9 of the updated Appendix, for multi-task DMLab-30 using all advantages does not substantially change the performance. However, for some continuous control problems we found the agents to learn less consistently without it. This was partly due to the fact that we share most parameters for the policy and value networks, and we effectively used the same weighting for the two losses across all tasks. This is now noted in the updated paper.
> >
> > It is possible that top-k can also help other algorithms, and as noted in the paper we did not invent it. However, it is unclear to us how it can easily be justified in all of those settings, whereas in MPO/V-MPO it corresponds to a specific choice of the improvement probability. Indeed, it may be of interest to note that the present formulation of the algorithm allows us to experiment with many other “tricks” depending on the setting that are not easily incorporated into other algorithms. In V-MPO a wide variety of choices are admissible. For instance, in Fig. 8 of the Appendix in the updated paper we have included an example of agents trained by *uniformly* weighting the top-k samples [similar to the Cross Entropy Method (CEM)]. Many such choices are possible, and different choices may be suitable for different applications.
> >
> > Previous work on multi-task DMLab-30 and Atari-57 also employed PopArt. More generally, PopArt is already known to benefit other algorithms such as DQN (van Hasselt et al., 2016) and IMPALA (Hessel et al., 2018), both in the single-task and multi-task settings, respectively; we would encourage wider adoption of this method by others. In the case of single-task training, it allows for more common hyperparameters to be shared across tasks when reward scales vary. Moreover, manual reward scaling of some kind is widespread (see, e.g., the discussion in https://openreview.net/forum?id=H1gdF34FvS), though not always made explicit. We use PopArt to automate this process and to keep hyperparameter settings similar across both multi-task and single-task experiments.
> >
> > “Minor: The mixture use of subscripts “old” and “target” is rather confusing, for example, in equation (5) and (15).”
> >
> > We wanted to convey that in the derivation, the V-MPO policy improvement step does not care in itself whether there is a target network, only that there is an old policy to be improved whose value function has been estimated. The target network is the practical instantiation of this policy iteration approach. In the updated paper we have clarified that the target network is a way to control the amount of data generated from the “old” parameters.

---

### Official Review · AnonReviewer1 · 2019-10-24
**Official Blind Review #1**

**Rating:** 6

**Review:**

The paper proposes an online variant of MPO, V-MPO. Compared to MPO, the main difference seems to be in the E-step. Instead of optimizing the non-parametric distribution towards a parameterized Q-function, V-MPO learns the V-function and updates the non-parametric distribution towards the advantages, which can be estimated on the samples of the last roll-outs based on the empirical returns and the learned V-function. Of course, updating towards (exponentiated and normalized) Q- or A-function does not make a difference. There are also some minor changes (which might still be crucial) such as an entropy constraint in the M-step and using only the top-k advantages during the E-Step (an option that was discussed in Abdolmaleki, et al. 2018a).
V-MPO is evaluated on DLM, ALE and two humanoid tasks from the DeepMind Control Suite. In most of these tasks V-MPO achieves returns that are to the best of my knowledge higher than any previously reported ones. However, the experiments also use a very large number of system interactions (in the order of billions).

Contribution / Significance:
I think that there will be relatively high interest in the paper due to the reported performances.
The technical contribution seems a bit incremental compared to MPO. Also, by learning a value function V-MPO gets closer to REPS. The submission lists the use of top-k samples and the M-step KL bound as the main differences to REPS. However, the former is not evaluated in the submission and the latter, albeit crucial, seems to be a relatively small modification. I do think that there are more differences to REPS, most important probably in the way of learning the value function and the corresponding differences in the derivations. However, I think that the differences abd similarities to MPO and REPS need to be discussed more thoroughly.

Soundness:
The derivation of V-MPO is relatively sound. The optimization of the KL constraints seems very approximate, although it seems to work well in practice.

Clarity:
I do not like the way the algorithm is presented. The submission specifies the complete loss function already at the beginning of the "Method"-Section and derives/motivates the individual terms in hindsight. The spaghetti-code like structure unnecessarily forces the reader to jump between pages or keeps the reader in the dark. I also do not like the "stop-gradient" notation which in my opinion puts the focus on low-level implementation details at the cost of not properly explaining what the optimization actually does. I think that paper is well-written in general, but the structure needs to be improved.

Experiments:
The evaluation clearly focuses on achieving the best performance, and does a good job in that regard.
However, a good evaluation should also help in understanding the mechanics of V-MPO. How does k (in top-k) affect the performance? How well are the constraints met during optimization? How does V-MPO compare to related on-policy methods (e.g. TRPO/PPO) on slightly more computational constrainted settings (eg rllab/mujoco with < 1e7 steps)?


Questions:
- Did you experiment with controlled entropy reduction akin to MORE, instead of using fixed
epsilon eta?

- Can you give a rough estimate of the computational time required to perform these experiments on a standard desktop pc? It really is difficult for me to even roughly estimate it.


Assessment:
Currently, I am leaning to accept because I think that V-MPO is overall a nice work. However I do think that submission needs to be revised. I mainly think that the structure needs to be improved and that V-MPO needs to better related to closely related work. I listed some additional experiments that would significantly improve the submission in my opinion.

**Experience Assessment:**

I have read many papers in this area.

**Review Assessment: Checking Correctness Of Derivations And Theory:**

I assessed the sensibility of the derivations and theory.

**Review Assessment: Checking Correctness Of Experiments:**

I assessed the sensibility of the experiments.

**Review Assessment: Thoroughness In Paper Reading:**

I read the paper at least twice and used my best judgement in assessing the paper.

---

> ### Author Response · Authors · 2019-11-13
> **Response to Reviewer #1 (1/2)**
>
> We thank the reviewer for their thoughtful questions and comments, and we appreciate their thorough understanding of the paper.
>
> We kept the discussion of similarities to MPO and REPS to a minimum due to space constraints in the initial submission, but in the revised paper the Related Works section has been expanded. It is additionally worth noting, though the reviewer is probably already aware, that in REPS the value function is a linear function of a learned feature representation whose features are learned by matching the feature distributions under the policy’s stationary state distribution. In contrast, in V-MPO the nonlinear neural network value function is learned directly from n-step returns. This is now noted in the Related Works section.
>
> “I do not like the way the algorithm is presented. The submission specifies the complete loss function already at the beginning of the "Method"-Section and derives/motivates the individual terms in hindsight. The spaghetti-code like structure unnecessarily forces the reader to jump between pages or keeps the reader in the dark. I also do not like the "stop-gradient" notation which in my opinion puts the focus on low-level implementation details at the cost of not properly explaining what the optimization actually does. I think that paper is well-written in general, but the structure needs to be improved.”
>
> Presenting the complete loss at the beginning was suggested to us by an earlier reader of the paper as a way of making the subsequent derivation easier to follow for those who prefer to see the loss upfront. However, not everyone appears to agree with this approach, and we have now moved the final loss to after the derivation. The updated paper also includes pseudocode for the algorithm, which is simpler than is perhaps suggested by the derivation!
>
> The stop-gradient is a way to implement coordinate descent while optimizing a single loss. We have attempted to clarify this in the updated paper as it is an important practical detail.
>
> “The evaluation clearly focuses on achieving the best performance, and does a good job in that regard. However, a good evaluation should also help in understanding the mechanics of V-MPO. How does k (in top-k) affect the performance? How well are the constraints met during optimization? How does V-MPO compare to related on-policy methods (e.g. TRPO/PPO) on slightly more computational constrained settings (eg rllab/mujoco with < 1e7 steps)?”
>
> In Figure 7 of the updated Appendix we now show an example of the different KL constraints during training. We note that not only are the constraints well satisfied, but they are saturated, which is desirable.
>
> We found the top-k selection to be more important in some tasks than others. In Figure 9 of the updated Appendix we show an example of multi-task DMLab-30 trained with all advantages included in the E-step, showing no appreciable difference in the final performance at 10B environment steps. In some continuous control tasks, however, agents learned less reliably without it. We therefore chose top-k so that this single setting could be applied to all experiments.
>
> We would also like to take this opportunity to note that top-k is not the only possible choice. For instance, in Fig. 8 of the updated Appendix we have included an example of agents trained by *uniformly* weighting the top-k samples. Many such choices are possible, and different choices may be suitable for different applications. In this light, top-k is just one choice that worked well across the experiments in this paper, but this may not be true in other settings.
>
> We acknowledge that in the data regime of < 1e7 steps mentioned by the reviewer, the current implementation and hyperparameters would not perform as well as TRPO/PPO. We note that even in multi-task DMLab-30 and Atari-57, it is evident that with the current set of hyperparameters V-MPO underperforms IMPALA early in training but eventually reaches substantially higher scores later.

---

> > ### Author Response · Authors · 2019-11-13
> > **Response to Reviewer #1 (2/2)**
> >
> > “Did you experiment with controlled entropy reduction akin to MORE, instead of using fixed
> > epsilon eta?”
> >
> > Model-Based Relative Entropy Stochastic Search
> > https://papers.nips.cc/paper/5672-model-based-relative-entropy-stochastic-search
> >
> > The main idea in MORE is to have both an upper bound on the KL divergence and a lower bound on the entropy, as a way of implicitly controlling the contribution of the shrinkage of the variance to the KL. This also happens in V-MPO in the continuous control case, but more explicitly in the M-step for continuous control, where we follow MPO in imposing separate upper bounds on both the mean and covariance components of the (parametric) KL divergence.
> >
> > “Can you give a rough estimate of the computational time required to perform these experiments on a standard desktop pc? It really is difficult for me to even roughly estimate it.”
> >
> > The algorithm and implementation were optimized for a distributed, high-throughput setup, and choices such as the batch size were chosen somewhat to take advantage of the available hardware. We are also cognizant of the fact that, unfortunately, this high data-regime is not available to everyone.
> >
> > However, in the past we have run similar experiments on commodity CPU (for the actor) and a multi-GPU setup (for the learner) available on many desktops, and in this case a comparable experiment might last several weeks. We were also happy to see another submission to ICLR, SEED RL (https://openreview.net/forum?id=rkgvXlrKwH), which details the time and cost of running similarly-scaled experiments on the Google Cloud platform.

---

> > > ### Comment · AnonReviewer1 · 2019-11-15
> > > **Thanks for the revision**
> > >
> > > Thank you for your replies and for updating the manuscript.
> > > I believe that the revision improved by having a better structure, related work and ablations.
> > >
> > > Some minor notes:
> > > - There is currently no fluent transition to Section "Policy Evaluation". Mentioning that Section 4.1 discusses the second point (of the enumeration in Section 4), whereas Section 4.2 discusses the third point would improve the flow.
> > >
> > > - The revision introduces a typo in Section 4.2.1: "this is corresponds to"
> > >
> > > - stop-gradient notation: It's a minor thing and it won't affect my decision. However, let me try one more time to convince you not to use such notation--because to me it has no benefits and just makes the paper a bit more complicated.
> > > I see that you want to show that you can implement the algorithm with a single loss function. However, when using non-standard mathematical notation that mixes equations with control flow (such as stop-gradients) you can probably show the same for any algorithm. An algorithm does not become simpler by reducing it to a single loss function--just as a program does not get simpler by writing the whole code in one line. It just gets harder to reader. But enough this.

---

> > > > ### Author Response · Authors · 2019-11-15
> > > > **Re: Thanks for the revision**
> > > >
> > > > We have added explicit references to Section 4.1 and 4.2 and now preview the derivation. This does indeed improve readability, thank you. We also fixed the typo, thanks for reading so carefully!
> > > >
> > > > We have now removed the stop gradient from the derivation of the M-step, as the algorithm itself does not require it. We think it is helpful to keep it in the specification of the final loss function, as it is directly tied to the actual implementation. We hope this presentation is more sensible.

---

### Official Review · AnonReviewer2 · 2019-10-25
**Official Blind Review #2**

**Rating:** 6

**Review:**

This paper proposes a new approximate policy iteration algorithm that is based on the previous method, called MPO, which formulates policy optimization (PO) as a probabilistic inference problem. The adaptation makes MPO become an on-policy method. One key modification made to MPO is to use advantage functions instead of Q-value function.

Overall, the paper follows an interesting topic in policy optimization as inference. It follows MPO to formulate PO as an inference problem using a similar principle. It proposes some extensions to MPO. The experiments show a lot of promising results. I have some concerns as follows.

- As V-MPO is developed based on MPO, however, the background on MPO is missing. This makes the reader hard to judge the novelty and difference of V-MPO against MPO. The discussion on the difference is very little.

- V-MPO extends MPO to on-policy setting, so the policy evaluation is key for this modification. Besides this modification, the E-step and M-step's derivations and objectives look quite similar to those of MPO. Can the authors comment on this?

- In addition, V-MPO is said to work for both discrete and continuous domains? It would be clearer if the authors discuss which parts in their algorithm enable this ability?

- It would also be great if the algorithmic description is included. The overall algorithm contains multiple optimization steps and hyperparameters, e.g. number of updates, how Lagrangian multiplier adapted etc..

- Continuous tasks: The comparisons with low sample-efficient approaches look unfair. Are there ablations that show performance comparisons of all when set with an equal level of samples?

- Can the author elaborate on the comment "These must be consistent between the maximum likelihood weights in Eq. 3 and the temperature loss in Eq. 4".



**Experience Assessment:**

I have read many papers in this area.

**Review Assessment: Checking Correctness Of Derivations And Theory:**

I assessed the sensibility of the derivations and theory.

**Review Assessment: Checking Correctness Of Experiments:**

I assessed the sensibility of the experiments.

**Review Assessment: Thoroughness In Paper Reading:**

I read the paper at least twice and used my best judgement in assessing the paper.

---

> ### Author Response · Authors · 2019-11-13
> **Response to Reviewer #2**
>
> We thank the reviewer for their thoughtful questions and comments, and we appreciate their thorough understanding of the paper.
>
> “As V-MPO is developed based on MPO, however, the background on MPO is missing. This makes the reader hard to judge the novelty and difference of V-MPO against MPO. The discussion on the difference is very little.”
>
> We kept this discussion to a minimum due to the space constraints in the initial submission, but in the revised paper the Related Works section has been expanded.
>
> “V-MPO extends MPO to on-policy setting, so the policy evaluation is key for this modification. Besides this modification, the E-step and M-step's derivations and objectives look quite similar to those of MPO. Can the authors comment on this?”
>
> This is true - to an extent. This is precisely why we chose the name V-MPO, to make it clear that conceptually, V-MPO arises from the same essential formalism as MPO. Demonstrating that this results in a highly competitive algorithm in practice is indeed the main contribution of this work.
>
> “In addition, V-MPO is said to work for both discrete and continuous domains? It would be clearer if the authors discuss which parts in their algorithm enable this ability?”
>
> There exist other algorithms, such as PPO, that work well for both discrete and continuous domains, so perhaps the question is, rather, why some do not. In general, any policy gradient algorithm using the likelihood-ratio trick to estimate gradients can, in theory, be applied to both continuous and discrete settings. However, two common problems are: 1) finding the right regularization of the policy that avoids collapse in high-dimensional continuous domains, and 2) dealing with the variance of the policy gradient estimate. V-MPO does not use the likelihood-ratio trick but separates the RL step from policy-fitting. The fitting step then does not require differentiating through observed returns, has low-variance, and is invariant to the scale of the returns (which makes finding the right regularization, e.g., the KL constraint here, somewhat easier).
>
> “It would also be great if the algorithmic description is included. The overall algorithm contains multiple optimization steps and hyperparameters, e.g. number of updates, how Lagrangian multiplier adapted etc.”
>
> The updated paper now includes pseudocode of the algorithm, which is simpler than is perhaps suggested by the derivation!
>
> “Continuous tasks: The comparisons with low sample-efficient approaches look unfair. Are there ablations that show performance comparisons of all when set with an equal level of samples?”
>
> Our implementation and hyperparameters were optimized to achieve the highest scores possible in the high-data regime; our goal was to show that these long-studied tasks have solutions with far higher returns than has previously been reported, and to demonstrate the existence of an algorithm, namely V-MPO, that can reliably reach them. We think this is interesting in and of itself.
>
> That said, we acknowledge that this on-policy version is optimized to reach high performance but does not outperform the other algorithms at the specified number of steps. Even in DMLab-30 and Atari-57, it is evident that with the current set of hyperparameters V-MPO underperforms IMPALA early on but eventually reaches substantially higher scores later.
>
> “Can the author elaborate on the comment "These must be consistent between the maximum likelihood weights in Eq. 3 and the temperature loss in Eq. 4".”
>
> In the derivation of V-MPO the same policy improvement probability p(I|s,a) enters in the weights and the temperature loss. Thus if the chosen p(I|s,a) includes top-k advantages only then it will appear in both the weights and the temperature loss. We have clarified this further in the updated paper.

---

### Public Comment · ~John_Schulman1 · 2019-09-29
**algorithm description would benefit from pseudocode; experiments should be apples-to-apples**

As written, the algorithm description involves a mixture of quantities we have access to p_{\theta_{old}}(s,a) and quantities that must be estimated, like A^{target}(s,a). Further, various important details, like the use of alternating optimization for the Lagrangian, are just mentioned in passing.

It'd be much easier to understand the algorithm if it were written out in pseudocode in terms of quantities we have numerical access to, such as advantage estimates and action probabilities. Also, the training loop for the constrained optimization problem should be written down more explicitly.

The experiments serve as a showcase that the method converges to good solutions, but it'd be more fair to compare to other methods wrt fixed sample complexity or computational complexity. As is, the comparisons of V-MPO with billions of timesteps vs other methods with millions aren't meaningful. The sample complexity comparison against impala+PBT also doesn't seem fair for obvious reasons.

---

> ### Author Response · Authors · 2019-10-02
> **Thanks for your comments! We will improve the presentation in revision / a few clarifications.**
>
> Thanks to the commenter for interest in our work!
>
> We will incorporate the commenter’s suggestions to improve the presentation of the algorithm in future revisions, including pseudocode of the algorithm. It is worth reiterating, however, that while the derivation of V-MPO involves a sequence of steps, the final algorithm consists only of gradient descent on the single loss function provided at the beginning of the Method section. In particular, there are no inner training loops or alternating optimizations in the implementation.
>
> Regarding the comparison to IMPALA-PBT, we note that the total number of environment frames used for IMPALA-PBT and V-MPO is the same. We did not use PBT with V-MPO to demonstrate that one can achieve superior performance with a single V-MPO agent, even when compared to a whole population of IMPALA agents. However, if we use PBT* then V-MPO achieves even higher scores in terms of both the max and mean across the population, see the Atari experiment in http://bit.ly/2nKTtPz. We will include this result in the updated paper.
>
> With respect to some of the other comparisons, particularly for continuous control tasks, as the commenter has rightly observed these figures are existence demonstrations - they only show the existence of much better solutions than has been reported in previous work, and the existence of an algorithm, V-MPO, that can reliably converge to these solutions. In Figure 7 of the Appendix, for example, we nearly double the previous best score for Gym Humanoid-V1 set by MPO. The other agents, whose sample complexity we have made explicit, are shown to provide a frame of reference for interpreting the final performance and we will make this clearer in revision.
>
> * Details of the Atari PBT experiment: All settings of the PBT experiment were the same as the one presented in the paper except the learning rates were also sampled log-uniformly from [8e-5, 3e-4) and epsilon_eta from [5e-2, 5e-1). Along with epsilon_alpha sampled log-uniformly from [1e-3, 1e-2) as in the original experiment, hyperparameters were evolved via copy and mutation operators roughly once every 4e8 environment frames.

---

### Decision · Program_Chairs · 2019-12-19

**Decision:**

Accept (Poster)

**Comment:**

This paper proposes an extension of MPO for on-policy reinforcement learning. The proposed method achieved promising results in a relatively hyper-parameter insensitive manner.

One concern of the reviewers is the lack of comparison with previous works, such as original MPO, which has been partially addressed by the authors in rebuttal. In addition, Blind Review #3 has some concerns with the fairness of the experimental comparison, though other reviews accept the comparison using standardized benchmark.

Overall, the paper proposes a promising extension of MPO; thus, I recommend it for acceptance.